# GRAPHENS: NEIGHBOR-AWARE EGO NETWORK SYNTHESIS FOR CLASS-IMBALANCED NODE CLASSIFICATION

**Joonhyung Park**[1*]**, Jaeyun Song**[1*]**, Eunho Yang**[1,2]
Korea Advanced Institute of Science and Technology (KAIST)[1], AITRICS[2]
{deepjoon, mercery, eunhoy}@kaist.ac.kr

## ABSTRACT

In many real-world node classification scenarios, nodes are highly class-imbalanced, where graph neural networks (GNNs) can be readily biased to major class instances. Albeit existing class imbalance approaches in other domains can alleviate this issue to some extent, they do not consider the impact of message passing between nodes. In this paper, we hypothesize that overfitting to the neighbor sets of minor class due to message passing is a major challenge for class-imbalanced node classification. To tackle this issue, we propose *GraphENS*, a novel augmentation method that synthesizes the whole ego network for minor class (minor node and its one-hop neighbors) by combining two different ego networks based on their similarity. Additionally, we introduce a saliency-based node mixing method to exploit the abundant class-generic attributes of other nodes while blocking the injection of class-specific features. Our approach consistently outperforms the baselines over multiple node classification benchmark datasets.

## 1 INTRODUCTION

Node classification on graphs has attracted significant attention as the importance of large-scale graphs analysis increases in various domains such as bioinformatics and commercial graphs to name a few (Perozzi et al., 2016; Hamilton et al., 2017; Ying et al., 2018; Mohammadrezaei et al., 2018). For example, in retail services, acquiring the qualitative node representations for items or customers is critical for improving the quality of recommendation systems (Perozzi et al., 2016; Ying et al., 2018). Detecting abnormal users in social networks, as another example, is also closely related to classifying the property of each node (Mohammadrezaei et al., 2018). Recently, Graph Neural Networks (GNNs) have demonstrated their effectiveness on learning node representations (Hamilton et al., 2017; Kipf & Welling, 2017; Velickovic et al.). However, the nodes in many real-world graphs are inherently class-imbalanced (Mohammadrezaei et al., 2018; Wang et al., 2020a), hence GNNs are prone to be biased toward major classes, as in general class-imbalance tasks. This bias forces networks to poorly classify the nodes of minor classes, resulting in destructive impacts and a large cost to their services.

While the peculiar characteristics of imbalanced node classification and specialized solutions suitable for it have hardly been investigated, applying generic imbalance handling methods (Chawla et al., 2002; Cui et al., 2019; Cao et al., 2019; Kang et al., 2020; Menon et al., 2021) directly to the graph domain has several non-trivial challenges. One of the distinct natures of graph data is that adjacent nodes are involved in constructing the representation of each node, which makes the model more confused to learn the unbiased representation of minor class nodes. Here, we hypothesize that it is in fact more serious to overfitting to neighbors of minor nodes than to overfitting to the node feature itself. This 'neighbor memorization' is the critical obstacle to naively adopt the class-imbalance approaches of other domains such as re-weighting and re-sampling used in image classification. Specifically, re-weighting approaches (Cui et al., 2019; Tan et al., 2020; Cao et al., 2019; Menon et al., 2021; Hong et al., 2021), applying penalties according to the number of data, simply assign large weight to minor nodes, hence there is no change in neighbor sets of minor nodes

---

[*]Equal contribution.

observed during training. Re-sampling methods (Chawla et al., 2002; Kang et al., 2020; Zhang et al., 2021; Wang et al., 2021), sampling data to balance the number of data for each class, are also vulnerable to overfit on minor nodes with their neighbors. Another challenge of the re-sampling method, especially for oversampling variants, is determining how to connect the newly sampled nodes to the original graph. For example, simply connecting an oversampled node with all neighbors of the original node will change the edges of neighbor nodes as well and hence significantly alter the message passing to the neighbors, which might impair their class semantics. To mitigate this issue, GraphSMOTE (Zhao et al., 2021) exploits edge predictor to decide the connectivity with the neighbors of two minor nodes used in oversampling. Nevertheless, GNNs trained with GraphSMOTE still suffer from neighbor memorization when the number of minor nodes is limited.

In this paper, we propose *GraphENS*, a novel augmentation approach that synthesizes the *whole* ego network for minor class (minor node and its one-hop neighbors) by interpolating two different ego networks in the data; to enlarge the limited neighbor views of minor instances in the data, our method combines the ego network of anchoring minor node with that of randomly selected node from all classes, where it interpolates ego networks based on KL-divergence between model predictions of ego networks in order to keep the semantics of minor classes. Synthesized ego networks are attached to the original graph to construct a class-balanced graph, and GNNs are trained with the enlarged graph. *GraphENS* enables the model to learn the minor class nodes with feasible neighbors by generating the virtual ego networks.

To further prevent the synthesis of deleterious ego networks, we introduce a saliency-based node mixing approach to generate the central node of ego network. Our method separates class-generic node features from class-specific node features by using saliency information of each feature and exploits only class-generic attributes to combine with node feature of anchoring minor node. We validate our method on various benchmark datasets including citation networks (Sen et al., 2008), and Amazon product co-purchasing networks (Shchur et al., 2018) with diverse architectures such as GCN (Kipf & Welling, 2017), GAT (Velickovic et al.), and GraphSAGE (Hamilton et al., 2017), and confirm that our approach consistently outperforms baseline methods over various settings.

In summary, our contribution is threefold:

- We demonstrate that in class-imbalanced node classification, GNNs severely overfit to neighbor sets of minor class nodes, rather than to minor nodes themselves. This 'neighbor memorization' problem becomes severe when the number of minor nodes is extremely small.

- Our method effectively alleviates the neighbor memorization problem in class-imbalanced graphs by synthesizing feasible ego networks based on the similarity between source ego networks. We also block the injection of harmful features in generating the mixed nodes using node feature saliency.

- Through extensive experiments, we show that our approach consistently outperforms baselines on multiple benchmark datasets including real-world imbalanced datasets. Even in highly imbalanced synthetic graphs, our method exhibits superior performance.

## 2 RELATED WORK AND PRELIMINARY

### 2.1 CLASS IMBALANCE PROBLEM

The goal of class-imbalance handling in classification is to construct an unbiased classifier to the label distribution of the training set. There are three main streams: loss modification, post-hoc correction, and re-sampling approaches. Loss modification methods alter the objective function to assign more weights (Japkowicz & Stephen, 2002; Cui et al., 2019) or margins (Tan et al., 2020; Cao et al., 2019; Menon et al., 2021) on minor classes. Post-hoc correction strategies (Kang et al., 2020; Tian et al., 2020; Menon et al., 2021; Hong et al., 2021) remedy logits to compensate minor classes in the inference. Re-sampling approaches augment minor class data by sampling strategies (Kang et al., 2020; Liu et al., 2020; Ren et al., 2020) or generation (Chawla et al., 2002; Kim et al., 2020a; Chu et al., 2020; Zhang et al., 2021; Wang et al., 2021). Among minor class generation approaches, SMOTE (Chawla et al., 2002) is a widely used method to mix minor data with the nearest data of the identical class. Synthesizing minor class data from data of other classes (Kim et al., 2020a; Chu

et al., 2020; Wang et al., 2021) is introduced to exploit the rich information of other classes. Kim et al. (2020a) produces new minor class data by taking gradient steps to translate major class data into minor class data. Wang et al. (2021) synthesizes minor class data by combining features of minor class data with feature displacements of other data. To extend these approaches to the graph domain, structural aspects of graph have to be considered when generating minor instances.

In node classification, imbalance handling works (Zhou et al., 2018; Wang et al., 2020b; Shi et al., 2020; Zhao et al., 2021; Qu et al., 2021) are proposed to exploit structural information in graphs. DR-GCN (Shi et al., 2020) produces the virtual minor nodes generated by additional conditional GAN and regularizes the features of virtual nodes close to adjacent nodes. GraphSMOTE (Zhao et al., 2021) generates synthetic minor nodes by interpolating two minor class nodes and a (pre-trained) edge predictor determines the connectivity of synthesized nodes between synthesized nodes and neighbors of two source minor nodes. ImGAGN (Qu et al., 2021) synthesizes minor nodes by interpolating features among whole minor nodes with the generated weight matrix. Then the synthesized nodes are connected to the original minor nodes if weights in the matrix are larger than a fixed threshold. As GraphSMOTE and ImGAGN only utilize nodes of the identical minor class to generate minor nodes, the sample diversity of synthesized nodes would be significantly constrained. Moreover, ImGAGN mainly targets binary classification and its extension to multi-class classification is non-trivial since an independent generator is required per each class. Compared to these approaches, our GraphENS utilizes whole nodes to synthesize minor nodes, thus our method outperforms baselines when the number of minor classes is low in Section 5.3 (Table 4).

## 2.2 GRAPH NEURAL NETWORKS FOR NODE CLASSIFICATION

We briefly introduce GNNs for node classification tasks. Let us first define graph $G(V, E)$ where $V$ is the set of nodes and $E$ is the set of edges between two nodes. Let $X \in \mathbb{R}^{|V| \times d}$ be the node features whose the $i$-th row represents the $d$-dimensional feature of the $i$-th node. $\mathcal{N}(v)$ is the set of adjacent nodes that are directly connected to $v$: $\{u \in V | \{u, v\} \in E\}$. In node classification, each node in the graph corresponds to a class $y \in \{1, \ldots, C\}$ where $C$ is the number of classes. In this paper, we consider several variants of GNNs that consists of three differentiable functions: 1) message function $m_l$, 2) permutation invariant message aggregation function $\phi_l$, and 3) node update function $h_l$. Let $x_v^{(l)}$ be the latent vector of node $v$ at layer $l$. To simplify notations for the recursive definition of GNNs, we use $x_v^{(0)}$ to denote the input node feature. At each GNN layer, node features are updated as $x_v^{(l+1)} = h_l(x_v^{(l)}, \phi_l(\{m_l(x_v^{(l)}, x_u^{(l)}, e_{v,u}) | u \in \mathcal{N}(v)\}))$ to consider adjacent node embeddings. By passing these aggregated node features to a linear classifier as input, we obtain the prediction $\hat{y} = f(x_v)$ for node $v$ where $\hat{y}_c = P(y = c | x_v)$. As a representative example, a layer of Graph Convolutional Network (GCN) (Kipf & Welling, 2017) is defined as $x_v^{(l+1)} = \Theta \sum_{u \in \mathcal{N}(v) \cup \{v\}} \frac{e_{v,u}}{\sqrt{\hat{d}_u \hat{d}_v}} x_u^{(l)}$ where $\hat{d}_v = 1 + \sum_{u \in \mathcal{N}(v)} e_{v,u}$ with $e_{v,u}$ is the edge weight of edge $\{u, v\} \in E$ and $\Theta$ is a filter parameters. There are several variants depending on how these key components are designed, but they are beyond scope of our work.

## 3 NEIGHBOR MEMORIZATION PROBLEM

In this section, we define and investigate the *neighbor memorization problem* uniquely appearing in the node classification task of a class-imbalanced graph. Conventional imbalance handling approaches such as re-weighting and oversampling enable the model to train in a class-balanced manner. However, these methods are vulnerable to overfit toward minor classes (Zhou et al., 2020) since they solely assign large weights on limited minor instances in the training phase. In the node classification task, the unique characteristics of this task, which have not been considered until now, should be considered: not only overfitting to the minor node feature $v_{minor}$, but also overfitting to $\mathcal{N}(v_{minor})$ can be an issue. That is, the model is exposed to highly restricted neighboring structures for minor classes. Here, since GNN models are based on message passing that receives information from (a number of) neighboring nodes, we hypothesize that the overfitting and memorizing $\mathcal{N}(v_{minor})$ can be more serious than overfitting to $v_{minor}$, and we experimentally verify that below.

**Experimental setup** We first conduct experiments to observe overfitting to minor classes on class-imbalanced graphs. GNNs are trained with two widely used methods - re-weighting and over-

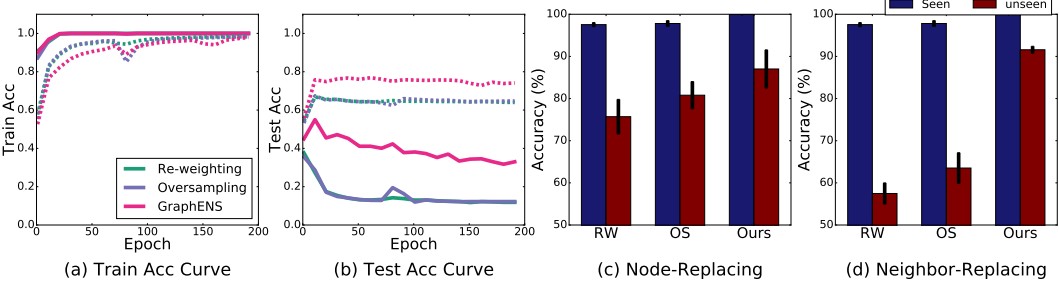

Figure 1: The results for overfitting to minor classes, node memorization, and neighbor memorization. Dash lines are learning curves including all classes and solid lines are learning curves for the minor class in (a) and (b). In (c), the blue/red bars indicate the accuracy of seen/unseen nodes with seen neighbors in the node-replacing experiments. In (d), the blue/red bars denote the accuracy of seen nodes with seen/unseen neighbors in the neighbor-replacing experiments. Note that RW and OS represent re-weighting and oversampling.

sampling on a node classification benchmark dataset, PubMed (Sen et al., 2008). We process the PubMed dataset to follow a long-tailed distribution by well-known construction rule in the vision domain (Cui et al., 2019). The imbalance ratio between the numbers of the most frequent class and the least frequent class is set as 100. We fix architecture as the 2-layer GraphSAGE having 256 hidden dimension and train models for 2000 epochs. We select the model by the validation accuracy.

**Overfitting to minor classes**    The learning curves of imbalance handling approaches are presented in Figure 1 (a) and (b). The solid lines and the dash lines are the curves of minor class accuracy and overall accuracy, respectively. We observe that the test accuracy of the minor class is much lower than the overall accuracy while the training accuracy of the minor class is extremely high. This result implies that the existing approaches are prone to memorize minor class nodes. Note that we only present the learning curves for the first 200 epochs to show the tendency of the early phase more clearly. We confirm the test accuracy differences between minor classes and all classes are almost maintained until the last epoch for both methods. The learning curve for entire epochs is in Appendix A.2 (Figure 4). In the following paragraph, we investigate more specifically whether this overfitting is mainly due to memorizing *node features* or their *neighbor structures*.

**Neighbor memorization problem**    To investigate how seriously GNN is prone to be overfitting to self node features or aggregated features from neighbors, we design two 'replacing' experiments. In the first replacing experiment, we scrutinize 'node memorization' where GNNs memorize seen minor node features excessively. We intentionally replace a seen minor node with an unseen minor node given a fixed seen neighbors (node-replacing experiment). Toward this, we randomly sample three nodes from the identical minor class: we sample $v_{anchor}$ (recipient), $v_{seen}$ (donator) from the training set and $v_{unseen}$ (donator) from the test set. We then replace the feature of $v_{anchor}$ with that of $v_{unseen}$ in the original graph and evaluate the classification accuracy of replaced $v_{anchor}$. That is, the replaced feature of $v_{anchor}$ is not exposed during training, but its neighbor view does. For comparison, we replace the feature of $v_{anchor}$ with of $v_{seen}$ as a baseline; it is the case of seen node features and neighbors but it has the artifact of grafting. We measure the accuracy for all minor test nodes and experiments are repeated 50 times. In Figure 1(c), the blue and red bars represent the results of $v_{anchor}$ with seen node features and $v_{anchor}$ with unseen node features, respectively.

Now, we conduct a similar experiment for 'neighbor memorization'. We would like to estimate how much GNN relies on seen topological structures. To this end, we measure the performance degradation when the seen neighbor set is replaced by an unseen neighbor set given a fixed seen self-node feature (neighbor-replacing experiment). We sample $v_{seen}$ (donator) and $v_{anchor}$ (recipient) from the training set and $v_{unseen}$ (recipient) from the test set as before, but $v_{anchor}$ now has a different role compared to the node-replacing experiment: it serves as providing a seen neighbor set. Specifically, we replace the feature of $v_{unseen}$ with that of $v_{seen}$, which represents the situation where a seen node is embedded into unseen neighbors. Then, we evaluate the classification accuracy of replaced $v_{unseen}$. For a baseline, the node feature of $v_{anchor}$ is replaced by that of $v_{seen}$ and measures the accuracy of replaced $v_{anchor}$; it is the case of seen node feature with a seen neighbor set. In Figure 1(d), the blue bar denotes the performance of seen node feature with seen neighbor set

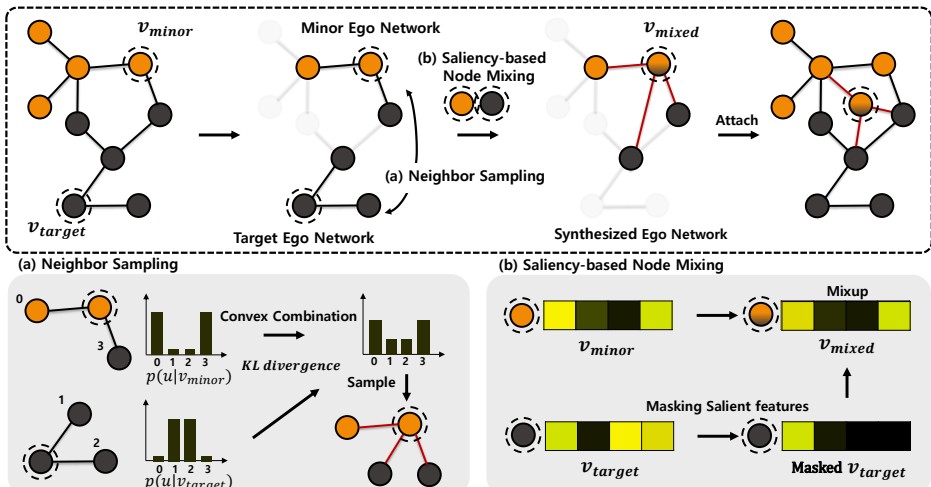

Figure 2: Overall pipeline of *GraphENS*. $v_{minor}$ is a node sampled from the minor class to be augmented (oversampled). $v_{target}$ is a node sampled from the entire class and exploited to synthesize the ego network.

and the red bar represents the performance of seen node feature with an unseen neighbor set. The overall procedure is described by Figure 8 in Appendix D.

We demonstrate that conventional imbalance handling approaches severely suffer from a neighbor memorization problem in Figure 1(c) and (d)[*]. The performance drop of conventional algorithms in the neighbor-replacing experiment is steeper than in the node-replacing setting. These results imply that the neighbor memorization problem is a critical obstacle in properly handling the class-imbalanced problem in node classification tasks.

From the learning curves and replacing experiments, we find that conventional approaches to handle imbalanced data largely rely on neighbor information. This neighbor memorization problem results in poor generalization performance on the minor class, so it must be appropriately addressed in class-imbalanced node classification. Recent work, GraphSMOTE (Zhao et al., 2021), connects a mixed node to neighbors of both the source node and target node in the same class, and then removes the unreliable edges by the feature similarity-based edge predictor. However, since GraphSMOTE only utilizes the neighbors of the seed nodes in the identical class, it still prone to memorize the neighbors of minor classes when the number of minor class nodes is extremely limited.

## 4    GRAPHENS

We now describe our novel class-imbalance handling strategy for node classification, *GraphENS* that synthesizes an ego network for generated minor nodes considering the prediction similarity of the ego network instances, in order to alleviate the neighbor memorization described above. *GraphENS* is mainly composed of two core components. Both components exploit the features from two nodes: the minor class node $v_{minor}$ and the target node $v_{target}$ from *entire classes*[*] to synthesize an ego network of minor class. The first component is to augment the neighbor set by combining neighbors of a minor node $v_{minor}$ with those of the target node $v_{target}$. To prevent the generation of harmful ego networks, *GraphENS* regulates the combination ratio depending on the KL divergence between ego networks centered on $v_{minor}$ and $v_{target}$, respectively (Section 4.1). The second component of *GraphENS* defines the node feature saliency for GNNs that represents the importance of each node feature in classifying the node property. Equipped with this node feature saliency, *GraphENS* mixes $v_{minor}$ and $v_{target}$ to generate a central node and combine with the ego network synthesized above (Section 4.2). This feature saliency information is utilized to filter out the class-specific node features in the node feature mixing process to preserve the semantics of the minor class. The overall procedure of *GraphENS* is described in Figure 2. Our full algorithm is also provided in the Appendix E (Algorithm 1). We discuss the details of each component in the following subsections.

---

[*]To reduce the negative effect of oversampling on message-passing, we make the message passing work only on incoming edges to the oversampled nodes.

[*]Detailed explanation and justification for this strategy for target node selection are discussed in Section 5.2.

### 4.1 Neighbor Sampling

Neighbor sampling is a method to determine which nodes to be connected to a new mixed node, $v_{mixed}$ (how to synthesize a mixed node is described in Section 4.2). The goal of neighbor sampling is to expose minor nodes to various environments to mitigate neighbor memorization problem while preventing the generation of harmful connections. To provide diverse neighbor sets, we construct 'adjacent node distribution' for a mixed node from two ego networks of $v_{minor}$ and $v_{target}$ and stochastically sample the neighbors from this distribution. At the same time, regarding the avoidance of detrimental connectivity, we devise a scheme that reflects $\mathcal{N}(v_{target})$ more in constructing the adjacent node distribution when the ego network of $v_{target}$ is more similar to that of $v_{minor}$. Note that the ego network $G_v$ of node $v$ consists of $\{\mathcal{N}_v \cup v\}$ nodes and edges directly connected to the central node $v$. The overall scheme of neighbor sampling is in Figure 2 (a).

**Constructing adjacent node distribution**  Now, we build the similarity-aware adjacent node distribution for $v_{mixed}$ by mixing adjacent node distributions of $v_{minor}$ and $v_{target}$. Here, the adjacency node distribution for a real node $v$, $p(u|v)$, is defined as $p(u|v) = \frac{1}{|\mathcal{N}(v)|}$ if $u \in V, \{u,v\} \in E$, and $p(u|v) = 0$ otherwise. To construct a 'probable' adjacent node distribution, the mixing ratio is determined based on the similarity between the ego networks of $G_{v_{minor}}$ and $G_{v_{target}}$. Toward this, we utilize logits $o_v = f(G)_v$ for a node $v$ given the full graph $G$ as input of GNN $f$ since GNN aggregates neighbor information with self node feature. It is worth noting that the logits can be obtained without additional cost (we can bring the model confidences at the previous epoch). To further reflect the neighbor information explicitly and avoid the overconfidence problem, we aggregate adjacent logits and utilize these logits to compute the similarity. Specifically, the aggregated logit of node $v$ is computed as $\hat{o}_v = \frac{1}{|\mathcal{N}(v)|+1} \sum_{u \in (\mathcal{N}(v) \cup v)} o_u$. Then, we calculate the KL-divergence between $\hat{o}_{minor}$ and $\hat{o}_{target}$ to estimate the distance between $G_{v_{minor}}$ and $G_{v_{target}}$: $\phi = KL(\sigma(\hat{o}_{minor})\|\sigma(\hat{o}_{target}))$ where $\sigma$ is a softmax function. Now, the normalized distance is used as the mixing ratio for the minor adjacent node distribution, and it is calculated as $\hat{\phi} = \frac{1}{1+e^{-\phi}}$. Finally, the adjacent node distribution for the synthesized $v_{mixed}$ is defined as

$$p(u|v_{mixed}) = \hat{\phi}\, p(u|v_{minor}) + (1 - \hat{\phi})\, p(u|v_{target}). \tag{1}$$

Note here that the mixing ratio of the target node decreases as the distance from the minor node increases and the mixing ratio of anchoring minor node is guaranteed to be at least $0.5$.

**Sampling neighbors**  Given the adjacent node distribution $p(u|v_{mixed})$ in Equation 1, we sample neighbors from this distribution without replacement. The number of neighbors is sampled from the degree distribution of a given graph to keep degree statistics with the original graph. Not only that, we make the message passing work only on the incoming edges to $v_{mixed}$ since undirected edges affect message passing in the original graph.

### 4.2 Saliency-based Node Mixing

We now describe how our method synthesizes new central node $v_{mixed}$ of $G_{v_{mixed}}$ by interpolating two nodes $v_{minor}$ and $v_{target}$. Note that our method does not restrict the target node to be from minor classes to utilize the abundant class-generic information of other classes. To extract only generic attributes independent of class, we first define the node feature saliency to identify the importance of each node feature. Armed with this node feature saliency, *GraphENS* synthesizes new minor node $v_{mixed}$ using minor node features and generic features of target node via convex combination.

**Feature saliency**  Albeit *GraphENS* does not depend on specific ways of calculating the importance of each node attribute, we simply adopt an approach that evaluates the importance of node features based on gradients. In particular, motivated by the works using the gradient of the loss function with respect to input features in image classifications (Simonyan et al., 2013; Shrikumar et al., 2017), we define the feature saliency as follows.

We first compute the gradient of classification loss $\mathcal{L}(G, \mathbf{y})$ with respect to the input node feature matrix $\frac{\partial \mathcal{L}}{\partial X} \in \mathbb{R}^{|V| \times d}$ via backpropagation. Then, the saliency value of $i$-th feature $s_{(v,i)}$ is obtained by the magnitude of the gradient regarding $i$-th attribute in node $v$ as $s_{(v,i)} = \left| \left[ \frac{\partial \mathcal{L}(G, \mathbf{y})}{\partial X} \right]_{(v,i)} \right|$. The

feature saliency vector $\mathcal{S}_v = [s_{(v,1)}, ..., s_{(v,d)}]^T$ is derived without any modification to the GNN architecture. Our gradient-based feature saliency can be derived without additional backpropagation as we simply exploit the gradient values computed from the previous training iteration.

**Node mixup**  We introduce our simple mixup-like augmentation to oversample the minor nodes without label mixing. Since we select target nodes $v_{target}$ from all classes unlike SMOTE (Chawla et al., 2002), filtering (or masking) out class-specific attributes of the target node is required to prevent the generation of noisy samples. To this end, we explicitly utilize the feature saliency information of the target node.

First, we determine the masking ratio $K\%$ using a normalized distance $\hat{\phi}$ (from Section 4.1). As $\hat{\phi}$ implies the distance between two ego networks centralized by $v_{minor}$ and $v_{major}$, we adaptively assign $K\%$ proportional to normalized distance $\hat{\phi}$; $K = k\hat{\phi}$ where $k$ is a hyperparameter. The intuition behind here is if target node $v_{target}$ is significantly differ from minor node $v_{minor}$, we mask more salient features of $v_{target}$. Specifically, given the feature saliency vector $\mathcal{S}_{v_{target}}$, a binary mask $M_K \in \mathbb{R}^d$, which masks the $K\%$ of node attributes to 0, is sampled from multinomial distribution by applying softmax to $\mathcal{S}_{v_{target}}$. Our final synthesized minor nodes $v_{mixed}$ can be formulated as follows given a random mixing ratio $\lambda \sim Beta(\alpha, \alpha)$:

$$v_{mixed} = (1 - \Lambda_K) \odot v_{minor} + \Lambda_K \odot v_{target}, \text{ where } \Lambda_K = \lambda \cdot M_K. \qquad (2)$$

We also evaluate our method but without using saliency on multiple datasets in Section 5.4 (Table 3).

## 5 EXPERIMENT

### 5.1 EXPERIMENT SETUP

**Datasets**  We validate *GraphENS* on five benchmark datasets: Cora, CiteSeer, PubMed for citation networks (Sen et al., 2008), AmazonPhoto and AmazonComputers for co-purchase graphs (Shchur et al., 2018). For citation networks, we follow splits in Yang et al. (2016). In supervised learning setting, we construct long-tailed citation networks following Cui et al. (2019) to validate models at a high imbalance ratio, which denotes the ratio between the most frequent class and the least frequent class. Nodes are removed until the class distribution follows a long-tailed distribution with keeping the connection in graphs at most. We sort the classes from major class to minor class and then remove nodes for each class started from major classes. When eliminating nodes, we remove nodes having low degrees and the corresponding edges of those nodes. We also evaluate imbalance handling methods on natural imbalanced datasets such as AmazonPhoto and AmazonComputers. To make validation/test sets balanced, we sample the same number of nodes from each class for validation/test sets. Then, the remaining nodes are assigned to the training set. We fix the least number of train nodes for all classes as 20. In semi-supervised setting, we set the imbalance ratio as 10. The detailed setup is provided in Appendix F.5.

**Baselines**  We test our methods over three architectures as GCN (Kipf & Welling, 2017), GAT (Velickovic et al.), and GraphSAGE (Hamilton et al., 2017). We adopt re-weight as cost-sensitive loss (Japkowicz & Stephen, 2002) inversely proportional to the number of class data. Oversampling is the approach that samples each class node until the number of each class data reaches the maximum number of class data. For oversampling, we duplicate the edges of the original node when adding an oversampled node to the original graph. cRT (Kang et al., 2020) and PC Softmax (Hong et al., 2021) are recent effective baselines for decoupling and post-hoc correction approaches, respectively. DR-GCN generates virtual minor nodes and forces virtual nodes to be similar to the neighbors of a source node. Although DR-GCN has an additional component to exploit entire unlabeled nodes for semi-supervised learning, we do not consider this component for a fair comparison. GraphSMOTE (Zhao et al., 2021) has two versions, depending on whether the predicted edges are discrete or continuous values. We adopt a discrete version of GraphSMOTE since it exhibits superior performance on multiple benchmark datasets. Implementation details of ours and description of evaluation protocol are deferred to Appendix F.

Table 1: Comparison of our method *GraphENS* with other baselines in extremely class-imbalanced settings (*Imbalance ratio:100*). We report the averaged accuracy, balanced accuracy, and F1-score with the standard errors for 5 repetitions on three benchmark datasets for node classification tasks.

| | Method | Cora-*LT* | | | CiteSeer-*LT* | | | PubMed-*LT* | | |
| --- | --- | --- | --- | --- | --- | --- | --- | --- | --- | --- |
| | | Acc. | bAcc. | F1 | Acc. | bAcc. | F1 | Acc. | bAcc. | F1 |
| GCN | Vanilla | 73.66 ±0.28 | 62.72 ±0.39 | 63.70 ±0.43 | 53.90 ±0.70 | 47.32 ±0.61 | 43.00 ±0.70 | 70.76 ±0.74 | 57.56 ±0.59 | 51.88 ±0.53 |
| | Re-Weight | 75.20 ±0.19 | 68.79 ±0.18 | 69.27 ±0.26 | 62.56 ±0.32 | 55.80 ±0.28 | 53.74 ±0.28 | 77.44 ±0.21 | 72.80 ±0.38 | 73.66 ±0.27 |
| | Oversampling | 77.44 ±0.09 | 70.73 ±0.10 | 72.40 ±0.11 | 62.78 ±0.37 | 56.01 ±0.35 | 53.99 ±0.37 | 76.70 ±0.48 | 68.49 ±0.28 | 69.50 ±0.38 |
| | cRT | 76.54 ±0.22 | 69.26 ±0.48 | 70.95 ±0.50 | 60.60 ±0.25 | 54.05 ±0.22 | 52.36 ±0.22 | 75.10 ±0.23 | 67.52 ±0.72 | 68.08 ±0.85 |
| | PC Softmax | 76.42 ±0.34 | 71.30 ±0.45 | 71.24 ±0.52 | 65.70 ±0.42 | **61.54** ±0.45 | **61.49** ±0.49 | 76.92 ±0.26 | **75.82** ±0.25 | 74.19 ±0.25 |
| | DR-GCN | 73.90 ±0.29 | 64.30 ±0.39 | 63.10 ±0.57 | 56.18 ±1.10 | 49.57 ±1.08 | 44.98 ±1.29 | 72.38 ±0.19 | 58.86 ±0.15 | 53.05 ±0.13 |
| | GraphSMOTE | 76.76 ±0.31 | 69.31 ±0.37 | 70.21 ±0.64 | 62.58 ±0.30 | 55.94 ±0.34 | 54.09 ±0.37 | 75.98 ±0.22 | 70.96 ±0.36 | 71.85 ±0.32 |
| | **GraphENS** | **77.76** ±0.09 | **72.94** ±0.15 | **73.13** ±0.11 | **66.92** ±0.21 | 60.19 ±0.21 | 58.67 ±0.25 | **78.12** ±0.06 | 74.13 ±0.22 | **74.58** ±0.13 |
| GAT | Vanilla | 73.60 ±0.26 | 62.75 ±0.37 | 63.53 ±0.35 | 56.76 ±0.39 | 50.15 ±0.34 | 46.59 ±0.44 | 71.26 ±0.77 | 58.86 ±0.82 | 54.91 ±1.12 |
| | Re-Weight | 77.26 ±0.09 | 70.97 ±0.11 | 71.37 ±0.33 | 63.54 ±0.39 | 56.98 ±0.35 | 55.30 ±0.42 | 78.14 ±0.09 | **75.80** ±0.14 | **76.07** ±0.12 |
| | Oversampling | 77.50 ±0.12 | 71.16 ±0.14 | 72.58 ±0.19 | 62.94 ±0.26 | 56.16 ±0.22 | 54.29 ±0.27 | 76.96 ±0.24 | 70.71 ±0.20 | 72.29 ±0.20 |
| | cRT | 76.74 ±0.30 | 69.19 ±0.52 | 70.60 ±0.57 | 60.80 ±0.53 | 54.18 ±0.55 | 52.33 ±0.70 | 73.76 ±0.23 | 65.69 ±0.25 | 66.40 ±0.27 |
| | PC Softmax | 76.62 ±0.15 | 73.14 ±0.19 | 72.69 ±0.19 | 62.20 ±0.42 | 59.49 ±0.40 | **59.51** ±0.51 | **78.70** ±0.60 | 75.40 ±0.72 | 75.96 ±0.62 |
| | DR-GCN | 74.42 ±0.55 | 64.17 ±1.00 | 64.20 ±0.67 | 56.74 ±0.74 | 50.02 ±0.75 | 45.82 ±1.03 | 71.52 ±0.25 | 59.18 ±1.06 | 54.879 ±2.33 |
| | GraphSMOTE | 76.92 ±0.31 | 70.03 ±0.51 | 70.47 ±0.55 | 64.04 ±0.38 | 57.33 ±0.39 | 55.43 ±0.49 | 77.12 ±0.49 | 73.59 ±1.16 | 74.40 ±0.72 |
| | **GraphENS** | **78.10** ±0.13 | **73.45** ±0.19 | **73.48** ±0.19 | **66.90** ±0.29 | **60.20** ±0.30 | 58.70 ±0.26 | 78.24 ±0.15 | 74.27 ±0.35 | 74.68 ±0.30 |
| GraphSAGE | Vanilla | 72.08 ±0.53 | 61.97 ±0.67 | 61.97 ±0.75 | 50.76 ±0.46 | 44.56 ±0.49 | 40.43 ±0.93 | 64.54 ±0.35 | 53.07 ±0.55 | 48.80 ±1.13 |
| | Re-Weight | 74.38 ±0.40 | 68.24 ±0.35 | 69.28 ±0.36 | 59.60 ±0.79 | 52.73 ±0.91 | 50.12 ±1.01 | 69.78 ±0.38 | 67.06 ±1.79 | 65.69 ±0.88 |
| | Oversampling | 74.88 ±0.64 | 66.20 ±1.38 | 66.36 ±1.11 | 58.64 ±1.17 | 51.76 ±1.17 | 49.42 ±1.44 | 69.36 ±0.12 | 62.15 ±0.83 | 62.46 ±0.74 |
| | cRT | 73.52 ±0.45 | 63.65 ±0.77 | 64.65 ±0.72 | 51.10 ±0.40 | 45.04 ±0.39 | 42.02 ±0.53 | 67.50 ±0.28 | 56.76 ±0.65 | 54.04 ±1.73 |
| | PC Softmax | 74.04 ±0.62 | 65.87 ±0.75 | 66.96 ±0.72 | 61.78 ±0.28 | 56.98 ±0.35 | **56.91** ±0.46 | 74.98 ±0.55 | 74.28 ±0.31 | 73.32 ±0.40 |
| | DR-GCN | 73.28 ±0.46 | 63.32 ±0.68 | 62.95 ±1.12 | 50.80 ±0.50 | 44.51 ±0.41 | 39.02 ±0.65 | 64.90 ±0.52 | 52.84 ±0.42 | 47.56 ±0.43 |
| | GraphSMOTE | 74.34 ±0.30 | 64.76 ±0.49 | 65.88 ±0.50 | 58.98 ±0.39 | 52.11 ±0.38 | 50.27 ±0.74 | 70.02 ±0.21 | 63.04 ±0.67 | 63.43 ±0.54 |
| | **GraphENS** | **77.26** ±0.13 | **70.07** ±0.28 | **70.25** ±0.31 | **63.98** ±0.38 | **57.33** ±0.42 | 55.23 ±0.43 | **79.60** ±0.19 | **74.90** ±0.49 | **75.83** ±0.43 |

Table 2: Results on AmazonPhoto and AmazonComputers in comparison to baselines. The experiment was repeated 5 times and standard errors are reported. Note that the Acc. is equal to bAcc. as the test sets are balanced.

| | Method | AmazonPhoto (*Imbalance ratio: 82*) | | AmazonComputers (*Imbalance ratio: 244*) | |
| --- | --- | --- | --- | --- | --- |
| | | Acc.(bAcc.) | F1 | Acc.(bAcc.) | F1 |
| GraphSAGE | Vanilla | 82.86 ±0.30 | 78.72 ±3.18 | 68.47 ±2.19 | 64.01 ±3.18 |
| | Re-Weight | 92.94 ±0.13 | 92.95 ±0.13 | 90.04 ±0.29 | 90.11 ±0.28 |
| | Oversampling | 92.46 ±0.47 | 92.47 ±0.48 | 89.79 ±0.16 | 89.85 ±0.17 |
| | cRT | 91.24 ±0.28 | 91.17 ±0.29 | 86.02 ±0.55 | 86.00 ±0.56 |
| | PC Softmax | 93.32 ±0.25 | 93.32 ±0.25 | 86.59 ±0.92 | 86.62 ±0.91 |
| | GraphSMOTE | 92.65 ±0.31 | 92.61 ±0.32 | 89.31 ±0.34 | 89.39 ±0.35 |
| | **GraphENS** | **93.82** ±0.13 | **93.81** ±0.12 | **91.94** ±0.17 | **91.94** ±0.17 |

Table 3: Ablation study. SM, NS, and PS denote saliency masking, neighbor sampling, and prediction similarity, respectively.

| | Method | CiteSeer-*semi* (*Imbalance ratio: 10*) | | | AmazonPhoto (*Imbalance ratio: 82*) | | |
| --- | --- | --- | --- | --- | --- | --- | --- |
| | | Acc | bAcc. | F1 | Acc. | bAcc. | F1 |
| GraphSAGE | GraphENS (*w/o SM,NS*) | 41.82 | 39.31 | 32.15 | 93.17 | 93.17 | 93.15 |
| | GraphENS (*w/o SM*) | 49.24 | 48.77 | 45.85 | 93.54 | 93.54 | 93.53 |
| | GraphENS (*w/o PS*) | 49.66 | 47.96 | 45.93 | 93.28 | 93.28 | 93.26 |
| | **GraphENS** | **51.12** | **48.91** | **46.78** | **93.82** | **93.82** | **93.81** |

## 5.2 TARGET NODE SELECTION

Our method utilizes the target node $v_{target}$ from all classes, not only from the identical minor class of $v_{minor}$. The intention of constructing the target node pool from entire classes is to guarantee sufficient diversity of synthesized ego networks. If the target nodes are restricted to a minor class at highly imbalanced scenarios, identical neighbor nodes would be redundantly duplicated. This issue makes it hard to mitigate the neighbor memorization problem. To demonstrate this hypothesis, we conduct an experiment and compare our design choice (all classes) to select a target from the identical class as the minor node $v_{minor}$. In the Figure 3, we confirm that exploiting entire classes for the target node achieves superior performances. From this design choice of the target node, our method can utilize the manifold assumption that 'similar predictions of neural networks indicate the close proximity in the manifold', which are commonly utilized in semi-supervised learning (Van Engelen & Hoos, 2020). We aim to enlarge and smooth the decision boundary of the minor class by interpolating the minor nodes and the target node of the entire classes (but excluding target-specific attributes using saliency) as investigated in Verma et al. (2019). Moreover, our method considers the prediction of the ego network (not a single node) to reflect the structural aspects of graph.

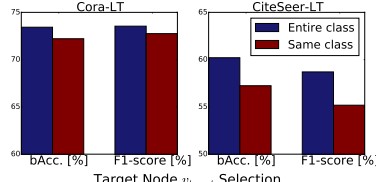

Figure 3: Target node experiments (GAT).

To select a target node, we sample a target class from the multinomial distribution with $(logN_1, logN_2, \dots, logN_C)$ where $N_i$ is the number of the $i$-th class data and $C$ is the number of classes. Then, we randomly select a target node from the training nodes of the sampled class.

Table 4: Comparison of our method *GraphENS* with other baselines in semi-supervised learning settings. We report the averaged accuracy, balanced accuracy, and F1-score with the standard errors for 5 repetitions.

| Method | Cora-*Semi* | | | CiteSeer-*Semi* | | | PubMed-*Semi* | | |
|---|---|---|---|---|---|---|---|---|---|
| | Acc. | bAcc. | F1 | Acc. | bAcc. | F1 | Acc. | bAcc. | F1 |
| Vanilla | 68.38 ±0.79 | 62.04 ±0.75 | 60.92 ±1.20 | 39.32 ±0.68 | 41.18 ±0.62 | 34.58 ±0.43 | 59.32 ±2.24 | 63.50 ±2.39 | 58.30 ±2.50 |
| Re-Weight | 71.06 ±1.13 | 65.08 ±1.43 | 65.09 ±1.59 | 38.10 ±1.23 | 39.87 ±1.17 | 32.13 ±1.44 | 68.10 ±2.47 | 69.82 ±2.02 | 67.70 ±2.30 |
| Oversampling | 65.20 ±1.15 | 60.61 ±0.95 | 59.57 ±1.71 | 43.56 ±3.62 | 42.27 ±3.08 | 36.68 ±4.81 | 64.94 ±3.24 | 68.48 ±2.13 | 63.79 ±3.56 |
| cRT | 68.96 ±0.60 | 62.65 ±1.00 | 61.74 ±1.43 | 41.42 ±0.42 | 42.17 ±0.44 | 36.38 ±0.62 | 58.38 ±2.31 | 63.10 ±2.73 | 57.95 ±2.60 |
| PC Softmax | 70.96 ±1.13 | 65.34 ±1.31 | 64.63 ±1.69 | 45.64 ±1.32 | 46.57 ±0.86 | 41.55 ±0.81 | 61.96 ±2.95 | 66.54 ±1.98 | 60.06 ±3.24 |
| DR-GCN | 66.84 ±1.92 | 59.94 ±2.57 | 58.21 ±3.94 | 42.52 ±1.36 | 41.16 ±1.39 | 35.89 ±1.97 | 59.64 ±2.13 | 64.32 ±2.17 | 59.09 ±2.21 |
| GraphSMOTE | 69.20 ±1.83 | 63.43 ±2.13 | 62.35 ±2.89 | 44.76 ±3.17 | 43.50 ±2.43 | 38.36 ±3.24 | 62.94 ±2.36 | 67.29 ±1.78 | 62.08 ±2.76 |
| **GraphENS** | **72.68** ±0.76 | **67.67** ±0.65 | **67.94** ±0.94 | **53.18** ±2.90 | **52.20** ±2.17 | **49.48** ±3.28 | **69.98** ±2.41 | **72.06** ±1.53 | **69.53** ±2.31 |

*(Left margin label: GCN)*

## 5.3 MAIN RESULTS

**Neighbor sampling mitigates neighbor memorization**   We investigate neighbor memorization over conventional algorithms in Section 3. We validate ours with respect to overfitting to minor classes, node memorization, and neighbor memorization. Although our training accuracy of the minor class is similar to re-weighting and oversampling, the test accuracy of our method is much higher than baselines, which indicates that our approach mitigates overfitting to minor classes in Figure 1 (a) and (b). Our method significantly decreases the performance degradation when a seen neighbor set is replaced by an unseen neighbor set, implying that our approach can generalize well for unseen neighbor views compared to baselines (Figure 1 (d)). Neighbor sampling thus effectively alleviates the neighbor memorization by exposing minor nodes to diverse circumstances.

**Supervised Learning**   We evaluate *GraphENS* on highly class-imbalanced situations as most real-world graphs. First, we test our method on three long-tailed citation networks: Cora-*LT*, CiteSeer-*LT*, and PubMed-*LT*. In Table 1, our method brings better imbalance handling performance in most cases compared to other baselines. To verify the effectiveness of our method on real-world imbalanced graphs, we also test GNNs trained with *GraphENS* on naturally class-imbalanced benchmark datasets without any modifications in graphs. Our method outperforms other contenders by significant margins in AmazonComputers (Table 2). Note that our method consistently exhibits higher accuracy and F1 score in other architectures (Table 6 in Appendix B.1).

**Semi-supervised learning**   To validate our approaches when labeled nodes are extremely restricted, we conduct experiments in a semi-supervised setting on citation networks (Table 4). Our approach significantly outperforms baselines for all datasets. Ours also exhibits consistent superior performances in other GNN architectures (Table 7). Note that since *GraphENS* exploits entire classes to synthesize minor nodes, our method can avoid the generation of redundant nodes when the number of minor nodes is extremely low. Hence, our method significantly improves the performance against existing oversampling-based approaches- GraphSMOTE, Oversampling. Ours brings further performance gain by utilizing unlabeled nodes to generate ego networks (Appendix B.2).

## 5.4 ABLATION STUDY

 We verify two key components of our method: 'prediction similarity' on neighbor sampling and 'saliency masking' on node mixing. To this end, we introduce baselines: **1)** '*GraphENS* w/o PS' determines prediction similarity uniformly random, **2)** 'w/o SM' mixes nodes without saliency masking ($M_K = 1$), and **3)** 'w/o SM,NS' generates mixed minor nodes without masking and connecting to neighbors of the target. We find that removing each module drops accuracy (Table 3). Thus, we believe each module effectively contributes to restraining the generation of harmful ego networks.

## 6 CONCLUSION

We investigated that existing imbalance handling algorithms suffer from neighbor memorization problem in class-imbalanced node classification. Thus, we proposed a novel augmentation method, *GraphENS*, to synthesize an ego network for minor classes with neighbor sampling and saliency-based node mixing. We verified that our method effectively mitigates neighbor memorization by synthesizing diverse but probable minor ego networks. *GraphENS* demonstrated its effectiveness in that it outperforms baselines over multiple benchmark datasets with various GNN architectures.

ACKNOWLEDGEMENT

This work was supported by Institute of Information & Communications Technology Planning & Evaluation (IITP) grant (No.2019-0-00075, Artificial Intelligence Graduate School Program (KAIST), No.2019-0-01371, Development of brain-inspired AI with human-like intelligence, No.2021-0-02068, Artificial Intelligence Innovation Hub) and the National Research Foundation of Korea (NRF) grants (No.2018R1A5A1059921) funded by the Korea government (MSIT). This work was also supported by Samsung Electronics Co., Ltd (No.IO201214-08133-01).

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

# A AUXILIARY RESULTS FOR NEIGHBOR MEMORIZATION PROBLEM

In this section, we present experimental results which are not reported in the Section 3 due to the space constraints.

## A.1 FURTHER INVESTIGATION FOR NEIGHBOR MEMORIZATION

In Section 3, we verify our hypothesis - easy to overfit to neighbor structures - on the basis of the most simple imbalance handling approaches. To further strengthen our claim, we intensively examine the neighbor memorization problem with other recent baselines over five benchmark datasets. As can be seen from Table 5, we observe that baselines consistently suffer from neighbor memorization issues in most cases. In addition, our approach shows significantly superior generalization performance than baselines on both node and neighbor replacing experiments except for node-replacing experiments on PubMed-*LT*.

Table 5: The results of node replacing (node memorization) and neighbor replacing (neighbor memorization) experiments on five benchmark datasets. We report the averaged accuracy for 5 repetitions.

| Method | Cora-*LT* | | CiteSeer-*LT* | | PubMed-*LT* | | AmazonPhoto | | AmazonComputers | |
|---|---|---|---|---|---|---|---|---|---|---|
| | Acc. | | Acc. | | Acc. | | Acc. | | Acc. | |
| (Seen / Unseen) | Node | Neighbor | Node | Neighbor | Node | Neighbor | Node | Neighbor | Node | Neighbor |
| Re-Weight | 96.80 / 89.74 | 96.80 / 83.94 | 99.10 / 81.73 | 99.10 / 88.01 | 95.00 / 86.40 | 95.00 / 79.58 | 96.80 / 96.71 | 96.80 / 89.10 | 98.43 / 98.06 | 98.43 / 96.56 |
| Oversampling | 98.10 / 89.96 | 98.10 / 84.67 | 99.26 / 83.09 | 99.26 / 89.01 | 93.62 / 85.76 | 93.62 / 78.71 | 96.75 / 96.37 | 96.75 / 89.45 | 98.77 / 98.49 | 98.77 / 96.85 |
| PC Softmax | 96.91 / 89.78 | 96.91 / 81.42 | 86.44 / 79.58 | 86.44 / 62.48 | 92.57 / **89.07** | 92.57 / 84.08 | 90.74 / 90.72 | 90.74 / 86.82 | 69.43 / 69.64 | 69.43 / 70.19 |
| GraphSMOTE | 96.78 / 89.35 | 96.78 / 83.99 | 94.39 / 75.48 | 94.39 / 85.06 | 94.24 / 85.94 | 94.24 / 77.42 | 90.69 / 90.54 | 90.69 / 86.22 | 96.66 / 96.12 | 96.66 / 96.54 |
| **GraphENS** | 97.94 / **90.48** | 97.94 / **91.06** | 98.84 / **87.52** | 98.84 / **92.99** | 96.43 / 87.42 | 96.43 / **89.19** | 97.73 / **97.20** | 97.73 / **91.28** | 99.45 / **99.13** | 99.45 / **98.48** |

We conducted experiments with 2-layer GCN having 256 hidden dimensions and adopted PC Softmax and GraphSMOTE as recent algorithms since PC Softmax and GraphSMOTE outperform cRT and DR-GCN in the main experiments, respectively. Note that for these experiments we group classes with less than 100 training nodes as minor classes.

## A.2 LEARNING CURVE OF THE TRAINING PHASE

To show the early trend of the training process, we only show the learning curve for the first 200 epochs in our main paper. We observe that the gap between training and test accuracy is almost maintained over entire epochs in Table 4.

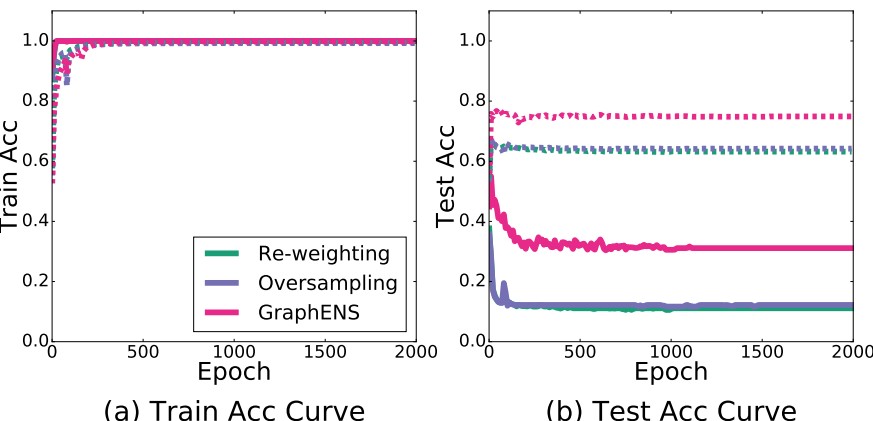

(a) Train Acc Curve          (b) Test Acc Curve

Figure 4: The learning curves over the training phase.

## B ADDITIONAL EXPERIMENTS ON MULTIPLE SETTINGS

### B.1 RESULTS OF AMAZONPHOTO AND AMAZONCOMPUTERS

Due to space constraints, we only present the experiment with GraphSAGE in the main paper, which exhibits the best vanilla performances, on two co-purchase graph datasets - AmazonPhoto, AmazonComputers. We also test our method with two different GNN architectures - GCN, GAT. As shown in Table 6, *GraphENS* shows the best performances over baselines in most cases. Detailed experimental settings are described in Section F.

Interestingly, we observe that cRT and PC Softmax show inferior performance than other algorithms on AmazonComputers (Table 6). We conjecture that these approaches fail to learn discriminative representations of minor classes due to message-passing in a *highly* imbalanced graph. Decoupling methods such as cRT and PC Softmax are proposed based on the observation that retraining the classifier with class-balanced sampling after training on imbalanced datasets show comparable performance with baselines (Kang et al., 2020). In other words, learning distinctive representation is not significantly affected by class imbalance. However, oversmoothing caused by message passing impedes learning distinctive representation of minor classes in highly imbalanced datasets. It is consistent with the observation that decoupling methods fail at the extremely high imbalance ratio (Ren et al., 2020).

Table 6: Accuracy/F1-score on AmazonPhoto and AmazonComputers benchmark datasets in comparison to other baselines. The experiment was performed five times and the averaged accuracy and F1-score with standard error are reported. Note that the accuracy is equal to balance accuracy since the test sets are balanced.

| | Method | AmazonPhoto (Imbalance ratio: 82) | | AmazonComputers (Imbalance ratio: 244) | |
|---|---|---|---|---|---|
| | | Acc.(bAcc.) | F1 | Acc.(bAcc.) | F1 |
| GCN | Vanilla | $81.91_{\pm0.25}$ | $77.50_{\pm0.28}$ | $58.42_{\pm0.29}$ | $51.21_{\pm0.39}$ |
| | Re-Weight | $92.73_{\pm0.12}$ | $92.72_{\pm0.13}$ | $90.65_{\pm0.25}$ | $90.70_{\pm0.23}$ |
| | Oversampling | $92.62_{\pm0.23}$ | $92.61_{\pm0.23}$ | $90.60_{\pm0.26}$ | $90.67_{\pm0.26}$ |
| | cRT | $90.43_{\pm0.16}$ | $90.20_{\pm0.21}$ | $78.06_{\pm1.30}$ | $77.60_{\pm1.35}$ |
| | PC Softmax | $91.05_{\pm0.18}$ | $91.09_{\pm0.18}$ | $75.89_{\pm3.92}$ | $74.87_{\pm4.56}$ |
| | GraphSMOTE | $92.58_{\pm0.11}$ | $92.58_{\pm0.11}$ | $91.00_{\pm0.21}$ | $91.05_{\pm0.21}$ |
| | **GraphENS** | $\mathbf{93.67}_{\pm0.18}$ | $\mathbf{93.67}_{\pm0.17}$ | $\mathbf{91.28}_{\pm0.18}$ | $\mathbf{91.28}_{\pm0.17}$ |
| GAT | Vanilla | $81.41_{\pm0.43}$ | $76.76_{\pm0.48}$ | $60.19_{\pm4.25}$ | $54.60_{\pm5.15}$ |
| | Re-Weight | $92.80_{\pm0.07}$ | $92.80_{\pm0.07}$ | $90.80_{\pm0.19}$ | $90.85_{\pm0.20}$ |
| | Oversampling | $92.69_{\pm0.18}$ | $92.68_{\pm0.19}$ | $90.54_{\pm0.20}$ | $90.56_{\pm0.22}$ |
| | cRT | $89.92_{\pm0.21}$ | $89.69_{\pm0.25}$ | $81.56_{\pm2.73}$ | $81.57_{\pm2.96}$ |
| | PC Softmax | $90.48_{\pm0.11}$ | $90.49_{\pm0.12}$ | $79.41_{\pm3.91}$ | $78.80_{\pm4.28}$ |
| | GraphSMOTE | $92.63_{\pm0.11}$ | $92.63_{\pm0.11}$ | $90.77_{\pm0.17}$ | $90.80_{\pm0.19}$ |
| | **GraphENS** | $\mathbf{93.10}_{\pm0.19}$ | $\mathbf{93.10}_{\pm0.20}$ | $\mathbf{91.11}_{\pm0.28}$ | $\mathbf{91.07}_{\pm0.28}$ |

### B.2 RESULTS OF SEMI-SUPERVISED LEARNING

In Section 5.3, we report the results of GCN on three citation networks - Cora, CiteSeer, PubMed. We here additionally offer the result table of two different GNN architectures - GAT, GraphSAGE. *GraphENS* outperforms baselines in most cases (Table 7).

To ensure that our method brings improvement for any data split (other than public split), we assess *GraphENS* on randomly selected train nodes while keeping the public split of validation/test nodes unmodified (Table 8). We confirm that the improvements of our method is not depending on a particular data split. We also attempt to extend *GraphENS* to utilize unlabeled nodes and validate this extended method. First, we regard unlabeled data as the $(C+1)$-th class ($C$ is the number of classes) and this class could be sampled when selecting a target node. Since the number of unlabeled data is large, the labeled class could be sampled less frequently if we utilize the number of unlabeled data in the multinomial distribution in Section 5.2. Thus, we treat the unlabeled class as the class having the same number of data with the least frequent class and sample a target class from the distribution with $(logN_1, logN_2, \ldots, logN_C, logN_{C+1})$ where $N_i$ is the number of the $i$-th class data. Note that we utilize model prediction on unlabeled nodes for neighbor sampling and saliency maps of source nodes for saliency masking. To keep class-specific information of source nodes,

Table 7: Comparison of our method *GraphENS* with other baselines in semi-supervised learning settings (***public split***). We report the averaged accuracy, balanced accuracy, and F1-score with the standard errors for 5 repetitions on three benchmark datasets for node classification tasks.

| | Method | Cora-*Semi* | | | CiteSeer-*Semi* | | | PubMed-*Semi* | | |
|---|---|---|---|---|---|---|---|---|---|---|
| | | Acc. | bAcc. | F1 | Acc. | bAcc. | F1 | Acc. | bAcc. | F1 |
| GAT | Vanilla | 69.56 ±0.69 | 63.57 ±1.07 | 63.27 ±1.11 | 42.86 ±0.86 | 43.28 ±1.12 | 39.30 ±1.28 | 66.78 ±1.81 | 70.80 ±1.36 | 66.46 ±1.72 |
| | Re-Weight | 73.24 ±1.57 | 67.68 ±1.62 | 67.36 ±1.75 | 52.58 ±0.79 | 51.92 ±0.49 | 50.22 ±0.48 | 67.28 ±1.71 | 70.63 ±1.01 | 67.07 ±1.65 |
| | Oversampling | 65.92 ±1.68 | 59.33 ±2.02 | 58.44 ±2.41 | 43.48 ±2.35 | 42.39 ±1.79 | 38.12 ±3.08 | 64.22 ±3.02 | 66.50 ±3.39 | 63.24 ±4.02 |
| | cRT | 69.68 ±0.80 | 63.44 ±0.82 | 62.98 ±1.12 | 46.40 ±1.93 | 45.87 ±1.27 | 43.71 ±1.66 | 65.72 ±2.49 | 69.82 ±1.88 | 65.26 ±2.40 |
| | PC Softmax | 71.06 ±1.37 | 65.89 ±1.63 | 65.35 ±1.91 | 53.40 ±1.21 | 52.66 ±1.10 | 50.58 ±1.83 | 68.88 ±2.18 | 71.65 ±1.40 | 68.67 ±2.01 |
| | DR-GCN | 68.84 ±0.71 | 62.87 ±0.86 | 62.20 ±1.25 | 46.06 ±1.52 | 46.26 ±1.37 | 42.05 ±2.08 | 65.48 ±2.76 | 69.31 ±2.19 | 65.03 ±2.51 |
| | GraphSMOTE | 70.20 ±1.05 | 64.99 ±1.51 | 64.59 ±1.67 | 46.42 ±2.27 | 46.63 ±1.74 | 41.92 ±2.26 | 66.42 ±3.06 | 68.86 ±2.00 | 65.26 ±3.70 |
| | **GraphENS** | **74.38** ±0.60 | **69.64** ±0.86 | **69.99** ±0.73 | **56.60** ±0.95 | **55.67** ±0.67 | **54.80** ±0.85 | **70.48** ±1.79 | **72.08** ±1.38 | **70.32** ±1.71 |
| GraphSAGE | Vanilla | 64.68 ±0.69 | 57.62 ±0.94 | 56.35 ±1.22 | 40.56 ±1.51 | 41.42 ±1.50 | 35.47 ±2.15 | 61.30 ±2.51 | 65.37 ±2.07 | 60.99 ±2.48 |
| | Re-Weight | 66.00 ±1.01 | 60.70 ±1.05 | 60.18 ±1.12 | 46.14 ±2.70 | 46.00 ±1.68 | 41.60 ±2.83 | 66.40 ±2.01 | 69.16 ±1.45 | 65.85 ±2.37 |
| | Oversampling | 62.04 ±1.18 | 53.66 ±1.45 | 50.55 ±3.10 | 38.14 ±1.51 | 38.00 ±0.75 | 31.57 ±1.04 | 64.74 ±3.13 | 67.71 ±2.45 | 64.72 ±3.02 |
| | cRT | 65.28 ±0.87 | 58.33 ±1.24 | 56.71 ±1.60 | 42.90 ±1.28 | 43.41 ±0.86 | 38.17 ±1.60 | 60.98 ±1.93 | 65.36 ±1.64 | 59.93 ±2.38 |
| | PC Softmax | 66.84 ±1.03 | 60.76 ±1.61 | 60.05 ±1.91 | 46.76 ±1.77 | 46.69 ±1.13 | 41.31 ±1.65 | 63.84 ±2.72 | 67.36 ±2.27 | 63.57 ±2.33 |
| | DR-GCN | 65.74 ±0.95 | 58.96 ±1.30 | 57.43 ±1.62 | 48.44 ±2.30 | 46.75 ±1.77 | 42.78 ±3.42 | 62.26 ±2.64 | 66.91 ±2.28 | 61.00 ±3.21 |
| | GraphSMOTE | 62.34 ±1.34 | 55.07 ±1.96 | 52.80 ±3.30 | 36.76 ±2.67 | 37.21 ±1.54 | 26.33 ±2.04 | 63.46 ±2.61 | 67.47 ±1.46 | 62.78 ±2.68 |
| | **GraphENS** | **71.18** ±0.72 | **66.50** ±0.75 | **66.56** ±0.63 | **53.48** ±1.50 | **52.57** ±0.73 | **50.56** ±1.45 | **69.44** ±2.16 | **70.67** ±1.64 | **69.59** ±2.01 |

exploitation of saliency maps of source nodes is required. In Table 8, the adoption of unlabeled data in *GraphENS* improves the performance significantly in most cases. We believe that our approach could exploit rich information even in unlabeled nodes to produce safe and diverse ego networks.

Table 8: Evaluation of *GraphENS* with other baselines in semi-supervised learning settings (***random split***). We report the averaged accuracy, balanced accuracy, and F1-score with the standard errors for 5 repetitions on three benchmark datasets for node classification tasks. † denotes the results when utilizing unlabeled nodes.

| | Method | Cora-*Semi* | | | CiteSeer-*Semi* | | | PubMed-*Semi* | | |
|---|---|---|---|---|---|---|---|---|---|---|
| | | Acc. | bAcc. | F1 | Acc. | bAcc. | F1 | Acc. | bAcc. | F1 |
| GCN | Vanilla | 63.98 ±1.32 | 58.28 ±1.59 | 55.97 ±2.33 | 40.76 ±2.68 | 40.54 ±2.22 | 35.11 ±3.33 | 57.48 ±2.67 | 62.35 ±1.41 | 55.27 ±3.58 |
| | Re-Weight | 67.56 ±1.32 | 63.05 ±1.10 | 61.43 ±2.59 | 42.90 ±2.97 | 41.42 ±3.06 | 35.22 ±4.15 | 66.98 ±2.36 | 69.54 ±1.83 | 66.35 ±2.32 |
| | Oversampling | 61.86 ±1.04 | 56.52 ±0.86 | 53.43 ±1.21 | 36.34 ±1.70 | 35.93 ±1.57 | 27.75 ±3.19 | 63.76 ±1.99 | 68.45 ±1.64 | 62.37 ±2.35 |
| | cRT | 63.94 ±1.14 | 57.16 ±1.49 | 54.45 ±2.23 | 41.16 ±2.73 | 41.78 ±2.36 | 36.08 ±3.24 | 57.34 ±2.93 | 62.21 ±2.12 | 55.39 ±3.63 |
| | PC Softmax | 69.20 ±1.02 | 64.68 ±1.76 | 63.80 ±2.05 | 45.24 ±2.85 | 44.34 ±2.33 | 40.38 ±2.86 | 64.78 ±0.75 | 66.89 ±1.37 | 64.82 ±0.74 |
| | CPGNN | 65.30 ±0.58 | 58.30 ±0.82 | 54.79 ±0.84 | 41.38 ±2.32 | 41.47 ±1.73 | 34.85 ±2.88 | 62.18 ±1.80 | 68.41 ±1.38 | 60.56 ±2.10 |
| | DR-GCN | 64.22 ±1.34 | 58.28 ±1.40 | 55.93 ±2.64 | 35.80 ±1.94 | 36.51 ±2.03 | 27.62 ±3.15 | 66.13 ±2.40 | 68.42 ±1.16 | 59.67 ±4.64 |
| | GraphSMOTE | 64.70 ±1.76 | 58.71 ±2.20 | 56.31 ±3.29 | 39.74 ±2.62 | 38.27 ±2.04 | 31.45 ±3.39 | 64.00 ±1.41 | 67.90 ±1.12 | 63.44 ±1.51 |
| | **GraphENS** | 69.78 ±1.01 | 65.01 ±1.67 | 64.69 ±1.60 | 52.42 ±1.09 | 49.55 ±1.18 | 46.27 ±1.54 | 68.54 ±1.15 | **70.68** ±1.22 | 68.19 ±0.93 |
| | **GraphENS†** | **70.76** ±0.86 | **66.78** ±1.31 | **66.20** ±1.54 | **53.58** ±1.26 | **51.69** ±1.03 | **49.42** ±1.71 | **69.54** ±2.32 | 70.38 ±1.97 | **69.00** ±2.07 |
| GAT | Vanilla | 67.80 ±1.30 | 64.08 ±1.44 | 63.02 ±1.66 | 43.42 ±1.69 | 42.39 ±2.02 | 38.30 ±2.13 | 66.16 ±1.75 | 70.37 ±1.16 | 65.67 ±1.69 |
| | Re-Weight | 69.40 ±1.42 | 65.14 ±1.92 | 63.90 ±2.43 | 50.56 ±1.72 | 47.83 ±1.67 | 45.28 ±2.45 | 68.94 ±2.04 | 71.57 ±1.33 | 68.78 ±1.84 |
| | Oversampling | 65.36 ±1.09 | 59.72 ±1.28 | 57.54 ±1.43 | 40.58 ±3.80 | 38.23 ±3.25 | 31.47 ±5.16 | 64.72 ±2.33 | 67.10 ±2.35 | 63.18 ±2.47 |
| | cRT | 67.56 ±1.65 | 62.10 ±2.27 | 61.15 ±2.61 | 45.58 ±1.89 | 43.45 ±2.26 | 40.15 ±2.36 | 65.96 ±2.94 | 70.10 ±2.17 | 65.27 ±3.02 |
| | PC Softmax | 69.30 ±1.17 | **66.05** ±1.47 | **65.06** ±1.61 | 50.08 ±2.13 | 48.61 ±2.15 | 47.19 ±2.09 | 68.42 ±1.16 | **71.83** ±0.86 | 68.28 ±1.06 |
| | DR-GCN | 66.76 ±1.77 | 61.47 ±2.43 | 59.95 ±3.07 | 41.28 ±3.14 | 40.71 ±2.37 | 33.96 ±3.66 | 66.96 ±1.80 | 69.02 ±1.52 | 66.11 ±2.10 |
| | GraphSMOTE | 68.23 ±1.31 | 63.67 ±1.67 | 62.50 ±2.15 | 45.26 ±2.71 | 43.42 ±3.06 | 39.46 ±3.65 | 65.16 ±1.43 | 69.32 ±0.71 | 64.39 ±1.58 |
| | **GraphENS** | **69.62** ±0.91 | 65.27 ±1.55 | 64.16 ±1.87 | 53.90 ±1.25 | 51.74 ±0.82 | 49.47 ±1.57 | **70.24** ±1.45 | 71.00 ±1.13 | **69.93** ±1.21 |
| | **GraphENS†** | 69.02 ±1.29 | 64.66 ±1.75 | 63.74 ±2.13 | **56.32** ±2.00 | **53.69** ±1.71 | **51.76** ±2.24 | 70.18 ±2.14 | 70.52 ±1.70 | 69.42 ±1.84 |
| GraphSAGE | Vanilla | 62.30 ±0.99 | 56.24 ±1.65 | 52.84 ±2.47 | 40.16 ±0.92 | 39.59 ±1.10 | 34.12 ±0.53 | 62.30 ±3.61 | 66.57 ±3.15 | 61.02 ±4.14 |
| | Re-Weight | 62.92 ±0.40 | 55.94 ±0.72 | 54.06 ±0.66 | 41.10 ±3.39 | 39.86 ±3.09 | 33.80 ±4.16 | 65.90 ±2.19 | **69.27** ±1.40 | 65.59 ±2.21 |
| | Oversampling | 61.40 ±0.83 | 52.99 ±1.23 | 49.57 ±1.63 | 39.04 ±2.00 | 38.89 ±1.99 | 32.17 ±2.88 | 59.26 ±2.26 | 63.59 ±2.55 | 58.06 ±2.78 |
| | cRT | 63.20 ±1.27 | 57.21 ±1.69 | 54.71 ±2.57 | 40.34 ±0.56 | 40.59 ±0.84 | 36.18 ±0.76 | 62.90 ±2.25 | 67.13 ±1.78 | 61.52 ±2.80 |
| | PC Softmax | 65.32 ±1.16 | 60.52 ±1.52 | 58.91 ±1.94 | 44.02 ±2.49 | 44.34 ±1.78 | 38.99 ±3.25 | 63.18 ±3.00 | 66.94 ±2.51 | 61.19 ±4.44 |
| | DR-GCN | 61.82 ±0.62 | 55.14 ±0.79 | 51.93 ±1.01 | 42.86 ±2.03 | 41.72 ±1.38 | 37.49 ±2.64 | 62.50 ±1.98 | 65.65 ±2.25 | 61.73 ±1.91 |
| | GraphSMOTE | 60.84 ±1.56 | 53.38 ±1.87 | 49.31 ±3.27 | 39.28 ±2.04 | 39.27 ±2.16 | 32.31 ±2.98 | 61.34 ±2.73 | 66.07 ±1.93 | 58.71 ±4.10 |
| | **GraphENS** | 65.84 ±0.74 | 62.29 ±1.69 | 61.52 ±1.79 | **51.12** ±1.93 | 48.91 ±1.94 | 46.78 ±2.02 | **67.62** ±1.77 | 68.27 ±1.26 | 67.37 ±1.60 |
| | **GraphENS†** | **67.72** ±1.04 | **64.18** ±1.31 | **63.14** ±1.47 | 50.92 ±2.13 | **49.94** ±1.60 | **47.06** ±2.23 | 67.48 ±1.19 | 68.04 ±0.76 | **67.64** ±0.99 |

## C    IN-DEPTH ANALYSIS

### C.1    CLASS-WISE PERFORMANCE ANALYSIS

To clearly present alleviation of overfitting to minor classes, we provide the test accuracy curves for each class. Note that PubMed-*LT* has three classes and the number of nodes for each class is as follows: ( Minor_1: 72, Minor_2: 726, Major: 7260 ). The experiment settings are identical to Section 3.

In Figure 5 (a), while the training accuracies for minor classes (Minor_1, Minor_2) are highly similar over entire methods, *GraphENS* exhibits significantly superior test accuracies for minor classes (Figure 5 (b,c)). Albeit our method slightly sacrifices the test accuracy of the major class, we confirmed that our approach substantially mitigates overfitting to the minor classes.

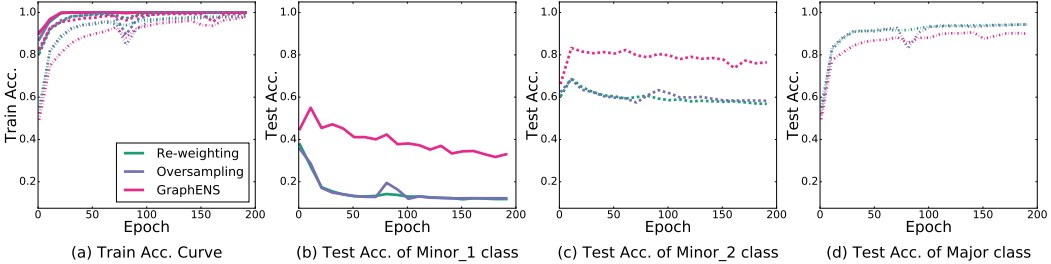

(a) Train Acc. Curve        (b) Test Acc. of Minor_1 class        (c) Test Acc. of Minor_2 class        (d) Test Acc. of Major class

Figure 5: The train accuracy curves for all methods and test accuracy curves for each class on PubMed-*LT* dataset. Solid lines represents Minor_1 class, dashed lines are Minor_2, and finely-dotted lines are Major class, respectively.

### C.2    VISUALIZATION OF THE HIDDEN REPRESENTATIONS

In this subsection, we shed light on why our algorithm works. Our method aims to enlarge and smooth the decision boundary of the minor class. First, to extend the boundary of the minor class, *GraphENS* utilizes the manifold assumption that "similar predictions of neural networks indicate the close proximity in the manifold" (Van Engelen & Hoos, 2020; Iscen et al., 2019) as a key inductive bias. By synthesizing the minor ego network using the target node of the entire class, our method widens the decision boundary effectively in that we interpolate the ego networks based on prediction similarity. In the vision domain, M2m (Kim et al., 2020b), one of the imbalance-handling methods, also translates major class data into minor class to prevent overfitting on minor classes exploiting our assumption implicitly. This extrapolation scheme for the minor class becomes more significant when the number of data is highly scarce.

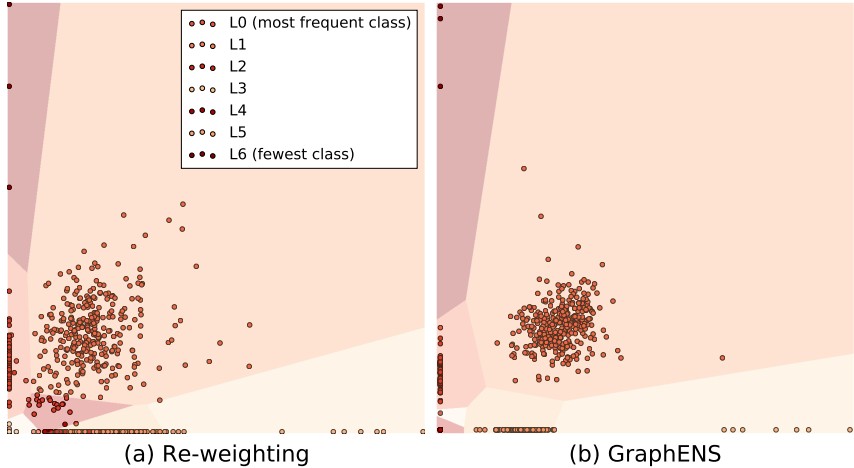

(a) Re-weighting                    (b) GraphENS

Figure 6: The decision boundaries of Re-weighting and our method are visualized by using 2D hidden representations of GNN.

Another advantage of our method for node classification is that the generalization performance of GNNs can be improved by smoothing the decision boundary. As investigated in Verma et al. (2019), convex combinations of node features encourage the smoothness of the boundary. To verify the smoothness and enlargement of minor classes, we visualize the decision boundaries of Re-weighting and our method by using 2D bottleneck hidden representations of GNN in Figure 6. Note that we conduct experiments on Cora-LT with 2 layers of GraphSAGE. Compared to Re-weighting, we observe that our boundaries are smoother and the fewest class occupies more region with significantly large margin. Therefore, along with the main results, this result implies that our method could improve generalization performance by enlarging the decision boundary of minor classes smoothly.

## C.3   OVERSAMPLING RATIO

We investigate the performance tendency according to oversampling scale, which determines the number of generated minor nodes. Specifically, the number of oversampled nodes is computed as oversampling ratio * (the number of nodes in the largest class - the number of nodes in the minor class)). In Figure 7, we observe that the performance decreases as the scale is (1) under 0.6 or (2) over 1.0. We conjecture that the induced bias of major classes is not sufficiently corrected at low oversampling scales (1). In the regime of (2), a high oversampling scale impedes the learning of major class representations, resulting in performance degradation. Thus, the oversampling scale between 0.8 and 1.0 is a preferable option and we adopted 1.0 as the scale in our main experiments, following conventional oversampling methods.

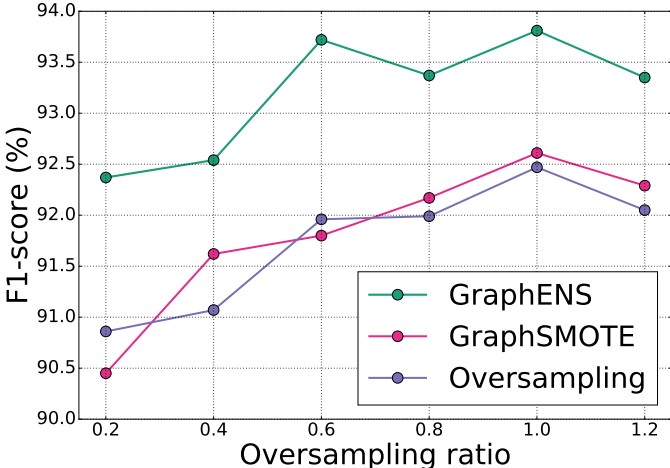

Figure 7: Performance analysis according to the oversampling ratio with AmazonPhoto dataset with Graph-SAGE.

# D   ILLUSTRATIONS OF REPLACING EXPERIMENTS

To facilitate understanding of node/neighbor replacing experiments, we describe the experiments with illustrations. In Figure 8, to observe node memorization, we replace the node feature of $v_{anchor}$ with $v_{seen}$ in (a) and $v_{anchor}$ with $v_{unseen}$ in (b). The performance gap between $v_{anchor}$ in (a) and (b) is computed. Similar to node-replacing experiments, to see neighbor memorization, we replace the node feature of $v_{seen}$ with $v_{anchor}$ in (c) and $v_{seen}$ with $v_{unseen}$ in (d). The accuracy difference between $v_{seen}$ in (c) and $v_{unseen}$ in (d) is calculated. The results of replacing experiments are in Figure 1.

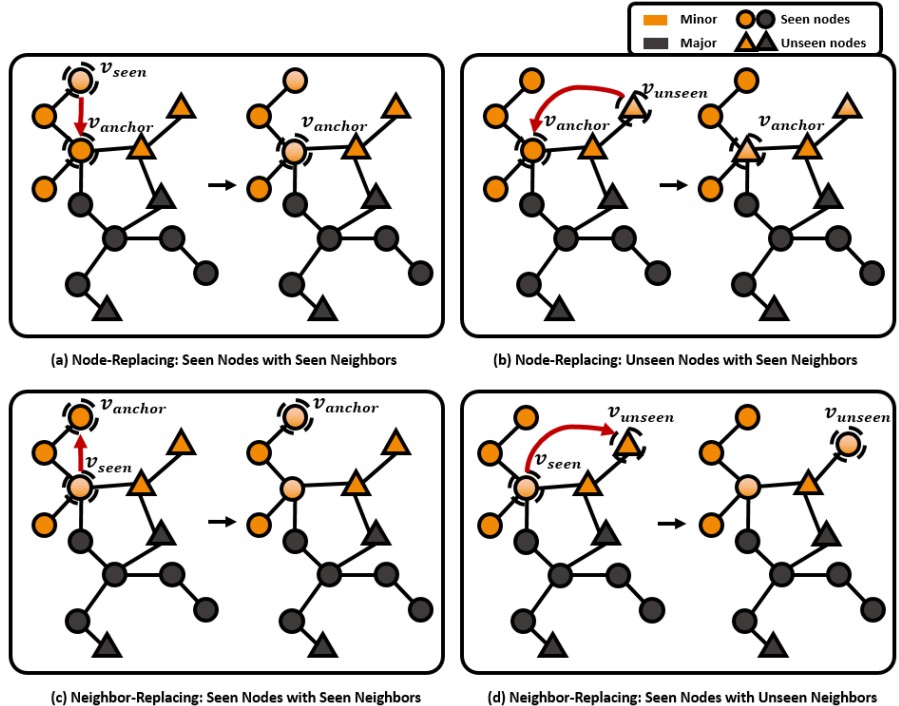

Figure 8: Overall pipeline of node-replacing and neighbor-replacing experiments.

# E  MAIN ALGORITHM

---

**Algorithm 1** *GraphENS*

---

1: **Input:** Dataset $(G(V, E), \mathbf{y})$, model $f_\theta$, class distribution $p_{class}(\mathcal{C})$, class-conditional node distribution $p_{node}(u|c)$, degree distribution $p_{degree}(\mathcal{D})$, number of classes $C$, number of samples to be oversampled $N_c^{os}$, feature masking hyperparameter $k$, learning rate $\eta$, temperature $\tau$

2: **Initialize:** Model $f_\theta$.

3: **for** $t = 1, 2, \ldots, T$ **do**

4:     $V_{new}, E_{new} \leftarrow V, E$

5:     **for** $c = 1, 2, \cdots, C$ **do**

6:         **for** $i = 1, 2, \cdots, N_c^{os}$ **do**

7:             $c_{target} \sim p_{class}(\mathcal{C})$                                  ▷ Sampling target class

8:             $v_{target} \sim p_{node}(u|c_{target})$

9:             $v_{minor} \sim p_{node}(u|c)$

10:           $\lambda \sim Beta(2, 2)$

11:           **if** $t \leq 10$ **then**                                         ▷ Warmup

12:              $v_{mixed} \leftarrow (1 - \lambda)v_{minor} + \lambda v_{target}$

13:              $v_{n_1}, v_{n_2}, \cdots, v_{n_{|\mathcal{N}(v_{minor})|}} \leftarrow$ Neighbors of $v_{minor}$

14:              $\hat{E}_{new} \leftarrow \{\{v_{n_1}, v_{mixed}\}, \{v_{n_2}, v_{mixed}\}, \cdots, \{v_{n_{|\mathcal{N}(v_{minor})|}}, v_{mixed}\}\}$

15:           **else**

16:              $\phi \leftarrow KL(\hat{o}_{v_{minor}}^{t-1} \| \hat{o}_{v_{target}}^{t-1})$

17:              $\hat{\phi} \leftarrow \frac{1}{1+e^{-\phi}}$

18:

19:              ▷ Saliency-based node mixing

20:              $K \leftarrow k\hat{\phi}$

21:              Compute $M_K$ from `multinomial distribution` using $\mathcal{S}^{t-1}, K$

22:              $\Lambda_K \leftarrow \lambda \cdot M_K$

23:              $v_{mixed} \leftarrow (1 - \Lambda_K) \odot v_{minor} + \Lambda_K \odot v_{target}$

24:

25:              ▷ Neighbor sampling

26:              $p(u|v_{mixed}) \leftarrow \hat{\phi}p(u|v_{minor}) + (1 - \hat{\phi})p(u|v_{target})$

27:              $r \sim p_{degree}(\mathcal{D})$

28:              $\{v_{n_1}, v_{n_2}, \cdots, v_{n_r}\} \sim p(u|v_{mixed})$

29:              $\hat{E}_{new} \leftarrow \{\{v_{n_1}, v_{mixed}\}, \{v_{n_2}, v_{mixed}\}, \cdots, \{v_{n_r}, v_{mixed}\}\}$

30:           **end if**

31:           $V_{new} \leftarrow V_{new} \cup \{v_{mixed}\}$

32:           $E_{new} \leftarrow E_{new} \cup \hat{E}_{new}$

33:         **end for**

34:     **end for**

35:     $\mathbf{o} \leftarrow f_\theta(V_{new}, E_{new})$

36:     $\mathcal{L}_{new} \leftarrow CrossEntropy(\mathbf{o}, \mathbf{y})$

37:     $\theta \leftarrow \theta - \eta(\nabla_\theta \mathcal{L}_{new})$

38:     **for** $v \in V$ **do**

39:         $\hat{o}_v^t \leftarrow \frac{1}{|\mathcal{N}(v)|+1} \sum_{u \in (\mathcal{N}(v) \cup v)} Softmax(\mathbf{o}_u/\tau)$                 ▷ Confidence aggregation

40:     **end for**

41:     $\mathcal{S}^t \leftarrow$ Computing feature saliency with the gradient of $\mathcal{L}_{new}$ (Equation. (2))

42: **end for**

43: **Output:** $f_\theta$

---

## F    Detailed Experiment Settings

In this section, we address all the details of experiments we conducted.

### F.1    Data statistics

Table 9: Data summary - Label distribution of training datasets [%]

| Dataset | Domain | $L_0$ | $L_1$ | $L_2$ | $L_3$ | $L_4$ | $L_5$ | $L_6$ | $L_7$ | $L_8$ | $L_9$ |
|---------|--------|-------|-------|-------|-------|-------|-------|-------|-------|-------|-------|
| Cora-*Semi* | Citation Network | 23.26 | 23.26 | 23.26 | 23.26 | 2.33 | 2.33 | 2.33 | - | - | - |
| Cora-*LT* | | 54.04 | 25.04 | 11.57 | 5.39 | 2.38 | 1.11 | 0.48 | - | - | - |
| CiteSeer-*Semi* | Citation Network | 30.30 | 30.30 | 30.30 | 3.03 | 3.03 | 3.03 | - | - | - | - |
| CiteSeer-*LT* | | 60.72 | 24.06 | 9.49 | 3.76 | 1.47 | 0.49 | - | - | - | - |
| PubMed-*Semi* | Citation Network | 83.33 | 8.33 | 8.33 | - | - | - | - | - | - | - |
| PubMed-*LT* | | 90.10 | 9.01 | 0.89 | - | - | - | - | - | - | - |
| AmazonPhoto | Co-Purchase Graph | 31.58 | 26.64 | 11.70 | 11.06 | 9.92 | 7.59 | 1.12 | 0.39 | - | - |
| AmazonComputers | Co-Purchase Graph | 44.26 | 17.07 | 16.94 | 10.35 | 4.95 | 2.45 | 1.96 | 1.49 | 0.34 | 0.18 |

### F.2    Architecture

We evaluate our method with three representative GNN architectures - GCN (Kipf & Welling, 2017), GAT (Velickovic et al.), and GraphSAGE (Hamilton et al., 2017). In this section, we describe the detailed architecture of each GNN. For GCN, our model consists of $l$ GCN layers with ReLU activation, followed by dropout Srivastava et al. (2014) with 0.5 dropping rate and a linear classifier. For GAT, our model comprises $l$ GAT layers, which involves an ELU activation Clevert et al. (2016), followed by dropout with 0.5 dropping rate and a linear classifier. We also adopt multi-head attention with 8 heads and apply dropout with 0.6 dropping rate to its attention. GraphSAGE is composed of $l$ SAGE layers with ReLU activation, followed by dropout with 0.5 dropping rate and a linear classifier. We search the architecture to maximize the validation accuracy for all algorithms. The search spaces of reported results are the number of layers $l \in \{1, 2, 3\}$ and hidden dimension $d \in \{64, 128, 256\}$.

### F.3    Evaluation Protocol

We adopt Adam (Kingma & Ba, 2015) optimizer with the initial learning rate as 0.01 and train GNNs for 2000 epochs. The best models are selected with validation accuracy. We design a scheduler as a learning rate is halved if there is no improvement on validation loss for 100 iterations. We choose weight decay to all convolutional layers except for a linear classifier with 0.0005. We select the best model using validation accuracy over the training phase.

### F.4    Implementation Details

In this subsection, we explain the hyperparameters of our experiments. For all datasets, we use $Beta(2, 2)$ distribution to sample $\lambda$. Feature masking hyperparameter $k$ and temperature $\tau$ are tuned among $\{1, 5, 10\}$ and $\{1, 2\}$, respectively. We employ warmup since our approach utilizes model confidence, so models need to be trained to some extent. The number of epochs for warmup is tuned among $\{1, 5\}$.

### F.5    The detailed setup for semi-supervised learning

We validate imbalance handling approaches on the citation networks such as Cora, CiteSeer, and PubMed. We follow the split in Yang et al. (2016) and set the imbalance ratio as 10. Specifically, for Cora and CiteSeer, three classes have only two labeled nodes while other classes have 20 labeled nodes since the number of labeled nodes in each class is 20 in Yang et al. (2016). For PubMed, only two classes have two labeled nodes as PubMed consists of three classes.

