# OpenReview forum: "GraphENS: Neighbor-Aware Ego Network Synthesis for Class-Imbalanced Node Classification"
_ICLR.cc/2022/Conference — ICLR 2022 Poster_

### Official Review · Reviewer_DE1Z · 2021-11-02

**Correctness:** 2
**Technical Novelty And Significance:** 2
**Empirical Novelty And Significance:** 2
**Recommendation:** 6
**Confidence:** 4

**Main Review:**

However, there are several major concerns in the paper.

1. A case study is conducted to prove the hypothesis that "we hypothesize that it is in fact more serious to overfitting to neighbors of minor nodes than to overfitting to the node feature itself". And this study fundamentally lays the foundation for the following part of this paper. However, there are several issues in this case study.

- (1.1) A benchmark dataset Pubmed is employed to conduct the case study. However, on different datasets, node representations calculated by GNNs may be influenced differently by the node-self features and the neighboring information, due to the distinct characteristics of datasets. Therefore, only employing one dataset here cannot demonstrate the existence of this phenomenon on different datasets with different distributions.

- (1.2) Two traditional baselines are utilized, namely re-weighting and oversampling approaches. However, many advanced approaches are proposed recently to deal with this problem. Only comparing with the most conventional approaches is less convincing to prove this hypothesis.

- (1.3) In the section of "Overfitting to minor classes", based on Figure 1, a claim is provided that "This result implies that the existing approaches are prone to memorize minor class nodes". However, this conclusion is not quite reasonable. Firstly, the case study only resorts to two conventional approaches, and they cannot represent all the existing approaches (for this problem). Secondly, from Figure 1, we can only observe that "the existing approaches are prone to overfit on minor classes, and the performance (on test) of them on minor classes is not good compared to the proposed GraphENS". Thus, the conclusion is over-claimed to some extent.

- (1.4) In summary of the case study, we can achieve a conclusion that: for a minor node, its neighboring information might provide a larger influence on the performance than its own features. However, it is hard to say: the neighboring information is the key factor to impair the performance of minor nodes.
Therefore, it seems that the case study cannot comprehensively provide the fundamental basis for the proposed model.


2. In the Introduction, a claim is presented that "Nevertheless, GNNs trained with GraphSMOTE still suffer from neighbor memorization when the number of minor nodes is limited". Why it still suffers from neighbor memorization? More details should be given.


3. The proposed model GraphENS contains two key components, namely neighbors sampling and saliency masking on features. From the main contribution of this paper, we can see that "the neighboring information is the key reason that impairs the performance of minor nodes". However, from the ablation study (Sect 5.4), we can see that, these two components generally have a comparable influence on the performance. The authors should give more explanations.


4. The illustration of Sect 4.1 is a bit confusing, especially the utilized notations. For example, is v_{minor} (or v_{target}) an example node or a group of nodes?


5. In the Experiments, in order to generate the imbalanced settings on the citation networks, for the minor classes, only two labeled nodes are available for training. The extreme scarcity of minor classes is not a reasonable setting and might import high bias or uncertainty into the experiments, which may damage the persuasiveness of the experiments accordingly, and even the case study.

**Summary Of The Paper:**

This paper investigates the problem of class-imbalanced node classification on graph. To address this, the authors first conduct a case study on a dataset to analyze the underlying reason for inferior performance on minor class nodes. Then a model named GraphENS, which consists of two key components, is proposed to deal with the imbalance issue. Experiments on several datasets show that the proposed model can outperform the baselines.

**Summary Of The Review:**

1. The case study is not convincing enough to lay the foundation for the model.
2. More details should be given for some confusing points.

---

> ### Author Response · Authors · 2021-11-16
> **Response to Reviewer DE1Z (3 / 3)**
>
> ### [Q5. In the imbalanced settings on the citation networks, for the minor classes, only two labeled nodes are available for training. The extreme scarcity of minor classes is not a reasonable setting.]
>
> To clarify the number of minor class nodes, we provide the table about the number of labeled nodes in the training set for each dataset. PubMed-*LT*, used in the case study, AmazonPhoto, and AmazonComputers have enough number of minor class nodes even though other datasets have highly scarce minor nodes as the reviewer mentioned. (Twenty nodes could be regarded as enough since twenty nodes are provided in the conventional semi-supervised learning setting on Cora, CiteSeer, and PubMed [1].)
> Thus, we could claim that our method shows superior performance over other contenders even in non-extreme scarcity cases (Table 1, 2).
>
> | **Dataset** (# of labeled nodes) | L0 | L1 | L2 | L3 | L4 | L5 | L6 | L7 | L8 | L9 |
> |---|:---:|:---:|:---:|:---:|:---:|:---:|:---:|:---:|:---:|:---:|
> | Cora-*Semi* | 20 | 20 | 20 | 20 | 2 | 2 | 2 | - | - | - |
> | CiteSeer-*Semi* | 20 | 20 | 20 | 2 | 2 | 2 | - | - | - | - |
> | PubMed-*Semi* | 20 | 2 | 2 | - | - | - | - | - | - | - |
> | Cora-*LT* | 341 | 158 | 73 | 34 | 15 | 7 | 3 | - | - | - |
> | CiteSeer-*LT* | 371 | 147 | 58 | 23 | 9 | 3 | - | - | - | - |
> | PubMed-*LT* | 7260 | 726 | **72** | - | - | - | - | - | - | - |
> | AmazonPhoto | 1630 | 1375 | 604 | 571 | 512 | 392 | 58 | **20** | - | - |
> | AmazonComputers | 4887 | 1885 | 1871 | 1143 | 547 | 271 | 216 | 165 | 37 | **20** |
>
>
> To further validate the effectiveness of our method when the number of minor class nodes is less scarce, we conducted extra experiments on the semi-supervised learning setting where each minor class has ten labeled nodes in the training set except for Cora-*Semi* (two labeled nodes are assigned for each minor class in the main paper). In Cora-*Semi*, each minor class has eight labeled nodes since some major classes in the original class have less than hundred nodes in the training set. (imbalance ratio is fixed as ten, so each major class should have hundred nodes) We can observe that our method outperforms baselines on Cora-*Semi* and Citeseer-*Semi* while the performance of ours is comparable to PC Softmax on PubMed-*Semi*. These results also support that our approach is also effective in non-extreme cases.
>
> | **Method** | **Cora-*Semi*** | **CiteSeer-*Semi*** | **PubMed-*Semi*** |
> |---|:---:|:---:|:---:|
> | (Arch:GCN, Imbalance ratio:10) | bAcc. \| F1 | bAcc. \| F1 | bAcc. \| F1 |
> | Vanilla | 69.87 \| 69.27 | 47.18 \| 42.44 | 74.52 \| 69.73 |
> | Re-weight | 74.40 \| 74.86 | 62.27 \| 61.67 | 75.89 \| 71.91 |
> | Oversampling | 74.36 \| 73.36 | 62.12 \| 61.53 | 73.50 \| 71.11 |
> | PC Softmax | 74.68 \| 74.62 | 58.85 \| 57.53 | **75.92** \| 73.01 |
> | GraphSMOTE | 73.91 \| 73.80 | 61.01 \| 60.54 | 73.89 \| 71.30 |
> | GraphENS | **77.84 \| 77.10** | **64.82 \| 64.29** | 75.57 \| **74.08** |
>
>
> [1] Zhilin Yang et al., Revisiting semi-supervised learning with graph embeddings, ICML 2016, Workshop.

---

> > ### Comment · Reviewer_DE1Z · 2021-12-01
> > **Change my score to weak accept**
> >
> > I appreciate the detailed responses from the authors, and they can address several of my concerns. I have read the comments from other reviewers, and have the following points.
> >
> > 1. In response 1/3, it is quite strange to employ GCN as the base GNN architecture. It is better to also apply the same GNN model as employed in the paper, i.e., GraphSAGE, which will be more convincing.
> >
> > 2. In response 3/3, I mean that there are only two labeled nodes in each minor class in PubMed-Semi, as illustrated in Appendix D.5. Besides, in some other datasets such as Cora-Semi and CiteSeer-Semi, only two labeled nodes are utilized in the minor classes. This extremely scarce data distribution (especially for the minor classes) would import high bias or uncertainty into the experiments, which may damage the persuasiveness of the experiments.
> >
> > 3. I agree with reviewer-KQwg that some notations are not clearly-illustrated, as I suggested in the reviews.
> >
> > Accordingly, I can change my score to weak accept, and recommend the authors to update the revision as suggested.

---

> > > ### Author Response · Authors · 2021-12-01
> > > **Thank you for the update and careful reading on our response**
> > >
> > > We thank the reviewer for checking out our response and providing positive feedback. We are happy to hear that our response helps to address the reviewer’s concerns. Below are answers to additional comments.
> > >
> > > ### [Q1. It is quite strange to employ GCN as the base GNN architecture. It is better to also apply the same GNN model (GraphSAGE) as employed in the paper.]
> > > We deliberately test the neighbor memorization problem in another GNN architecture in this experiment to show that the problem is not restricted to the specific architecture. In fact, we also confirmed that our method consistently exhibits significantly superior performances even for GraphSage in neighbor replacing experiments. Following the reviewer’s suggestions, we will include both results in our final paper. We believe these additional results will strengthen our paper. We want to thank the reviewer again for valuable suggestions.
> > >
> > > ---
> > >
> > > ### [Q2. The extremely scarce data distribution would import high bias or uncertainty.]
> > >
> > > We conducted experiments under the data-scare setting following the highly related work, GraphSMOTE [1]. However, we agree that extremely scarce data distribution would import high bias or uncertainty. Thus, we conducted additional experiments with enough minor nodes ($\geq 8$) under the identical imbalance ratio and confirmed that our approach consistently shows superior performance over baselines in response 3/3. We will add these extra experimental results in the final revision.
> > >
> > > [1] Tianxiang Zhao, et al. “GraphSMOTE: Imbalanced Node Classification on Graphs with Graph Neural Networks.” WSDM 2021.
> > >
> > > ---
> > >
> > > ### [Q3. Some notations are not clearly-illustrated.]
> > > As we responded to Reviewer KQwg, we did not re-define the terminology in the previous revision to avoid further misunderstanding, but we promise we will carefully revise the notations in a more clear way in the final revision.
> > >
> > > ---
> > >
> > > We will keep your advice in mind and carefully revise our paper including the above suggestions.

---

> ### Author Response · Authors · 2021-11-16
> **Response to Reviewer DE1Z (2 / 3)**
>
> ### [Q1.4. In summary of the case study, we can conclude that for a minor node, its neighboring information might provide a larger influence on the performance than its own features. It is hard to say that the neighboring information is the key factor to impair the performance of minor nodes.]
>
> As in general classification tasks, it is natural to think that (node) features are the main targets for overfitting in the node classification task of graph.  However, as shown in our experiments, and as mentioned by the reviewer, the structure/features of neighbors in the graph can be more important factors in overfitting than the node features themselves. (In fact, even without comparison with node features, the neighbor information shows a large effect by itself: Average performance drop of 10.10% for all baselines over 5 node classification benchmark datasets (Q.1.1. Q1.2). Therefore, we believe that it is not a logical leap to say that the influence of neighborhood information is the key factor. However, we will revise it to be more clear: “one of the major factors”. If the concern mentioned by the reviewer is not about this, please let us know.
>
> ---
>
> ### [Q2. Show that GraphSMOTE still suffers from neighbor memorization.]
>
> GraphSMOTE generates a mixed node by interpolating two nodes from the same minor class. Then, mixed nodes can only be connected to the neighbors of seed minor nodes. Hence, the synthesized nodes are continuously exposed to the restricted neighbor views of the minor class.
> To prove the above statement, we compare ours with GraphSMOTE on neighbor-replacing experiments over five node classification benchmark datasets. We confirmed that GraphSMOTE consistently exhibits inferior generalization performance compared to ours. This result supports that GraphSMOTE still suffers from neighbor memorization. We will add this discussion and results regarding GraphSMOTE in our revision.
>
>
> | Unseen neighbor | Cora-*LT* | CiteSeer-*LT* | PubMed-*LT* | AmazonPhoto | AmazonComputers |
> |---|:---:|:---:|:---:|:---:|:---:|
> | (Arch:GCN) | Acc. | Acc. | Acc. | Acc. | Acc. |
> | GraphSMOTE | 83.99 ± 0.29 | 85.06 ± 0.99 | 77.42 ± 0.75 | 86.22 ± 0.57 | 96.54 ± 0.28 |
> | GraphENS | **91.06 ± 0.42** | **92.99 ± 0.24** | **89.19 ± 0.56** | **91.28 ± 0.79** | **98.48 ± 0.15** |
>
> ---
>
> ### [Q3.  From the ablation study (Sect 5.4), these two components (neighbor sampling and saliency node mixup) generally have a comparable influence on the performance, but the main contribution of this paper is “the neighboring information is the key reason that impairs the performance of minor nodes”. The authors should give more explanations.]
>
> The ablation study (Section 5.4) verifies two key design choices - ‘‘prediction similarity (PS)’ in neighbor sampling and ‘saliency masking (SM)’ in saliency node mixup. That is, current ablation studies do not mainly target to directly compare the contribution of neighbor sampling and saliency node mixup.
>
> To address the reviewer's concerns, we conduct additional ablation experiments on five node classification benchmark datasets. We alternatively remove each component (neighbor sampling and saliency node mixup) from our method and evaluate the performance.
>
> | **Method** | **Cora-LT** | **CiteSeer-LT** | **PubMed-LT** | **AmazonPhoto** | **AmazonComputers** |
> |---|:---:|:---:|:---:|:---:|:---:|
> | (Arch:GCN) | bAcc. \| F1 | bAcc. \|  F1 | bAcc. \|  F1 | bAcc. \| F1 | bAcc. \|  F1 |
> | GraphENS (*w/o neighbor sampling*) | 69.01 \| 69.58 | 56.76 \| 55.02 | 71.31 \| 71.87 | 92.66 \| 92.64 | 90.69 \| 90.70 |
> | GraphENS (*w/o saliency node mixup*) | 69.90 \| 69.80 | 57.45 \| 55.49 | 73.63 \| 74.32 | 93.01 \| 93.00 | 90.93 \| 90.92 |
> | GraphENS | **72.94 \| 73.13** | **60.19 \| 58.67** | **74.13 \| 74.58** | **93.67 \| 93.67** | **91.28 \| 91.28** |
>
> As shown in the above table,  ‘w/o neighbor sampling’ consistently shows inferior results than ‘w/o saliency node mixup’ for all datasets which implies that neighboring information is the influential factor in classifying nodes. We thank the reviewer for the thoughtful comments.
>
> ---
>
> ### [Q4. The illustration of Sect 4.1 is a bit confusing, especially the utilized notations.]
>
> We apologize for the confusing notations. Both $v_{minor}$ and $v_{target}$ are example nodes. $v_{minor}$ is a node sampled from the minor class to be augmented (oversampled). $v_{target}$ is a node sampled from the entire class and exploited to synthesize the ego network. We will clarify this to avoid confusion in the revision.

---

> ### Author Response · Authors · 2021-11-16
> **Response to Reviewer DE1Z (1 / 3)**
>
> We sincerely appreciate your time and effort to review our paper. We address the reviewer’s concerns below.
>
> ### [Q1.1, Q1.2. Show the case study with other baselines over other datasets.]
>
> We intentionally considered the most basic methods of handling class imbalance for clarity, since we believe that our hypothesis - easy to overfit to neighbors rather than the node itself - does not depend greatly on the method. However, taking into account the comments by the reviewer, we intensively examine our hypothesis with other recent baselines over five benchmark datasets. In these new experiments, we observed that baselines consistently suffer from node and neighbor memorization issues, and confirmed that neighbor memorization is more serious in most cases. In addition, our approach shows significantly superior generalization performance than baselines on both node and neighbor replacing experiments except for node-replacing experiments on AmazonPhoto.
>
> Note that we conducted experiments with another GNN network- GCN and adopted PC Softmax and GraphSMOTE as recent algorithms since PC Softmax and GraphSMOTE outperform cRT and DR-GCN in the main experiments, respectively. We will happily report the results for these baselines(cRT, DR-GCN) if the reviewer hopes to see them.
>
> | **Cora-*LT*** | seen node | unseen node | seen neighbor | unseen neighbor |
> |---|:---:|:---:|:---:|:---:|
> | (Arch:GCN) | Acc. | Acc. | Acc. | Acc. |
> | Re-Weight | 96.80 ± 0.42 | 89.74 ± 0.56 | 96.80 ± 0.42 | 83.94 ± 0.39 |
> | Oversampling | 98.10 ± 0.28 | 89.96 ± 0.67 | 98.10 ± 0.28 | 84.67 ± 0.35 |
> | PC Softmax | 96.91 ± 0.27 | 89.78 ± 0.64 | 96.91 ± 0.27 | 81.42 ± 0.51 |
> | GraphSMOTE | 96.78 ± 0.26 | 89.35 ± 0.44 | 96.78 ± 0.26 | 83.99 ± 0.29 |
> | GraphENS | 97.94 ± 0.37 | **90.48 ± 0.37** | 97.94 ± 0.37 | **91.06 ± 0.42** |
>
> | **CiteSeer-*LT*** | seen node | unseen node | seen neighbor | unseen neighbor |
> |---|:---:|:---:|:---:|:---:|
> | (Arch:GCN) | Acc. | Acc. | Acc. | Acc. |
> | Re-Weight | 99.10 ± 0.13 | 81.73 ± 0.64 | 99.10 ± 0.13 | 88.01 ± 0.24 |
> | Oversampling | 99.26 ± 0.11 | 83.09 ± 0.60 | 99.26 ± 0.11 | 89.01 ± 0.26 |
> | PC Softmax | 86.44 ± 1.04 | 79.58 ± 0.47 | 86.44 ± 1.04 | 62.48 ± 0.63 |
> | GraphSMOTE | 94.39 ± 0.55 | 75.48 ± 1.46 | 94.39 ± 0.55 | 85.06 ± 0.99 |
> | GraphENS | 98.84 ± 0.13 | **87.52 ± 0.58** | 98.84 ± 0.13 | **92.99 ± 0.24** |
>
> | **PubMed-*LT*** | seen node | unseen node | seen neighbor | unseen neighbor |
> |---|:---:|:---:|:---:|:---:|
> | (Arch:GCN) | Acc. | Acc. | Acc. | Acc. |
> | Re-Weight | 95.00 ± 0.58 | 86.40 ± 0.88 | 95.00 ± 0.58 | 79.58 ± 0.56 |
> | Oversampling | 93.62 ± 1.17 | 85.76 ± 1.36 | 93.62 ± 1.17 | 78.71 ± 0.67 |
> | PC Softmax | 92.57 ± 0.57 | **89.07 ± 0.66** | 92.57 ± 0.57 | 84.08 ± 0.67 |
> | GraphSMOTE | 94.24 ± 0.75 | 85.94 ± 1.27 | 94.24 ± 0.75 | 77.42 ± 0.75 |
> | GraphENS | 96.43 ± 0.56 | 87.42 ± 0.56 | 96.43 ± 0.56 | **89.19 ± 0.56** |
>
> | **AmazonPhoto** | seen node | unseen node | seen neighbor | unseen neighbor |
> |---|:---:|:---:|:---:|:---:|
> | (Arch:GCN) | Acc. | Acc. | Acc. | Acc. |
> | Re-Weight | 96.80 ± 0.59 | 96.71 ± 0.60 | 96.80 ± 0.13 | 89.10 ± 0.80 |
> | Oversampling | 96.75 ± 0.72 | 96.37 ± 0.78 | 96.75 ± 0.72 | 89.45 ± 0.63 |
> | PC Softmax | 90.74 ± 0.57 | 90.72 ± 0.58 | 90.74 ± 0.57 | 86.82 ± 0.51 |
> | GraphSMOTE | 90.69 ± 0.37 | 90.54 ± 0.40 | 90.69 ± 0.37 | 86.22 ± 0.57 |
> | GraphENS | 97.73 ± 0.51 | **97.20 ± 0.67** | 97.73 ± 0.51 | **91.28 ± 0.79** |
>
> | **AmazonComputers** | seen node | unseen node | seen neighbor | unseen neighbor |
> |---|:---:|:---:|:---:|:---:|
> | (Arch:GCN) | Acc. | Acc. | Acc. | Acc. |
> | Re-Weight | 98.43 ± 0.22 | 98.06 ± 0.30 | 98.43 ± 0.22 | 96.56 ± 0.20 |
> | Oversampling | 98.77 ± 0.20 | 98.49 ± 0.21 | 98.77 ± 0.20 | 96.85 ± 0.22 |
> | PC Softmax | 69.43 ± 6.57 | 69.64 ± 6.54 | 69.43 ± 6.57 | 70.19 ± 5.52 |
> | GraphSMOTE | 96.66 ± 0.44 | 96.12 ± 0.50 | 96.66 ± 0.44 | 96.54 ± 0.28 |
> | GraphENS | 99.45 ± 0.18 | **99.13 ± 0.24** | 99.45 ± 0.18 | **98.48 ± 0.15** |
>
> ---
>
> ### [Q1.3. "This result implies that the existing approaches are prone to memorize minor class nodes" is not quite reasonable in that the authors conduct memorization experiments for two conventional methods. Secondly, we can only observe that "the existing approaches are prone to overfit on minor classes, and the performance of them on minor classes is not good".]
>
> As mentioned in Q1.2 above, we observed that many baselines including recent approaches suffer both node and neighbor memorization, so in the revision, we will carefully highlight that “many baselines are prone to memorize minor class nodes”. Regarding the second point, we basically used ‘memorization’ in the same sense as ‘overfitting’ (that is, it does not generalize well to unseen data), as is often the case in several existing studies including [1, 2].
>
> [1] Arpit, Devansh, et al. “A Closer Look at Memorization in Deep Networks” ICML 2017.
>
> [2] Zhang, Chiyuan, et al. “Understanding deep learning requires rethinking generalization” ICLR 2017.

---

### Official Review · Reviewer_ACEE · 2021-11-03

**Correctness:** 3
**Technical Novelty And Significance:** 3
**Empirical Novelty And Significance:** 3
**Recommendation:** 8
**Confidence:** 4

**Main Review:**

Strong points:

1. The proposed model is intuitive and technically sound. Neighbour sampling is important in graphs given the neighborhood memorization problem.

2. I like the preliminary analysis on the importance of neighbour sampling, putting the neighbor memorization problem in a quantitative manner (i.e. section 3)

3. Experiments are extensive with strong empirical results.

I only have one main concern:

The target node is picked from a distribution based on the log of the number of nodes in the class, but not much rationale is offered on why this particular distribution is chosen. Is there any guiding principle? Would the model benefit from more elaborately designed distributions?



**Summary Of The Paper:**

The paper proposes an imbalanced classification strategy for GNNs. Unlike traditional imbalanced classification, nodes on a graph are dependent on its neighbors, and simply over-sampling/re-weighting the minor class instances would not work. In particular, the authors recognize and analyze the severity of the "neighbor memorization" problem, which is identified as the key cause of overfitting in GNNs.
The proposed method alleviates the neighbor memorization problem by synthesizing ego networks. Extensive experiments are conducted to evaluate the proposed method.

**Summary Of The Review:**

The paper is well written and technical sound, with strong results.

---

> ### Author Response · Authors · 2021-11-16
> **Response to Reviewer ACEE**
>
> We thank the reviewer for the appreciation and valuable comments.
>
> ### [Q.The target node is picked from a distribution based on the log of the number of nodes in the class. Is there any guiding principle? ]
>
> As the reviewer mentioned, a target node class is sampled from the multinomial distribution that is proportional to *log* of the number of nodes in each class. We adopt this distribution to enhance the probability of minor classes being selected. Specifically, if we use multinomial distribution based on the number of nodes, minor classes are hardly selected in highly imbalanced scenarios.
>
> To validate our sampling strategy, we conduct comparative experiments in which we sample a target class from a distribution that does not use a logarithm. In the table below, we confirmed that sampling distribution with *log* consistently outperforms the sampling distribution without using *log*.
> Following the reviewer’s suggestion, we will explore more elaborated sampling distributions and add this comparison to our paper within this rebuttal period. Thank you for your constructive comments.
>
> | **Method** | **Cora-LT** | **CiteSeer-LT** | **PubMed-LT** |
> |---|:---:|:---:|:---:|
> | (Arch:GCN) | bAcc. \| F1 | bAcc. \| F1 | bAcc. \| F1 |
> | GraphENS (*w/o log*) | 72.81 \| 72.62 | 59.55 \| 58.04 | 71.54 \| 72.32 |
> | GraphENS | **72.94 \| 73.13** | **60.19 \| 58.67** | **74.13 \| 74.58** |

---

> ### Author Response · Authors · 2021-11-22
> **Second Response to Reviewer ACEE**
>
> As the reviewer suggested, we explored another sampling distribution to improve the performance. We design the **class-wise sampling distribution** encouraging to sample more target nodes from other classes as the number of nodes in a minor class gets fewer. Specifically, $\left(\frac{N_{minor}}{N_{max}}\right)^{\delta}$ is the probability to select the same class for target node ($p_{same}$) and the remainder ($1-p_{same}$) is uniformly distributed to other classes. Note that $N_{minor}$ and $N_{max}$ are the numbers of nodes in the minor class and the largest class respectively and $\delta$ is a hyperparameter. The intuition behind here is that minor classes could obtain additional gain via the convex combination with **other** classes compared to major classes. As investigated in [1], albeit this sampling rule brings marginal performance degradation to major classes, it would be beneficial in effectively expanding the boundary of minor classes.
>
> To validate a new sampling strategy, we conduct experiments on a long-tailed dataset and compare the new strategy with our original sampling strategy. We observe that the class-wise distribution is significantly effective on PubMed-*LT*, but shows inferior or comparable performance to the original strategy on Cora-*LT* and CiteSeer-*LT*. This implies that there is a small room for performance improvement by designing the elaborate distribution. We will leave it as future work.
>
> | **Method** | **Cora-*LT*** | **CiteSeer-*LT*** | **PubMed-*LT*** |
> |---|:---:|:---:|:---:|
> | (Arch:GCN) | bAcc. \| F1 | bAcc. \| F1 | bAcc. \| F1 |
> | GraphENS | **72.94 \| 73.13** | **60.19 \| 58.67** | 74.13 \| 74.58 |
> | GraphENS w/ class-wise dist. | 72.07 \| 72.46 | 60.04 \| 58.50 | **75.66 \| 76.15** |
>
> [1] Kim, Jaehyung, et al. "M2m: Imbalanced classification via major-to-minor translation." CVPR 2020.

---

> > ### Comment · Reviewer_ACEE · 2021-12-09
> > **Thanks for additional experiments**
> >
> > I think the experiment with regards to the new distribution is interesting. Overall this is a good paper.

---

### Official Review · Reviewer_6ykZ · 2021-11-04

**Correctness:** 3
**Technical Novelty And Significance:** 3
**Empirical Novelty And Significance:** 4
**Recommendation:** 6
**Confidence:** 4

**Main Review:**

The paper has multiple strong points and few weak points.

Strong points:

1. The paper motivates the problem very well. The experiments in Section 3 show the gap that existing / conventional approaches for class imbalance problem fail to address for graphs. Thus, it shows very well why graph-specific solutions for class imbalance problem are needed.

2. The related work and baselines used for experiments are quite exhaustive to the best of my knowledge.

3. Experimental results are quite promising. They clearly show the merit of the algorithm and its components for imbalanced node classification.

4. The source code is attached (I have not run it though).


Weak Points / Need Clarifications:

1. Though I appreciated the motivation and problem formulation of the paper, the proposed algorithm GraphENS is very heuristic in nature. Different steps of the algorithm, i.e., Neighbor Smapling and Node Mixing do not have a strong objective. I agree that these steps are simple and also intuitive. But what (and how) advantage it is gaining for node classification objective is not clear. It would have been great if some objective / cost function can be formulated and these steps could have been deduced by solving that. At least, authors should give more justification on their need for node classification.

2. Table 1 shows improvement on the manually imbalanced citation datasets. I am curious to see the performance of GraphENS on the original citation datasets. It would be great if authors can compare the performance using GCN, GAT and GraphSAGE with / without GraphENS.

3. What is the training, validation and test size for supervised and semi-supervised learning setup? I also think that "Supervised Learning" task is also a semi-supervised one as it is a GNN based node-level task on a graph. Can authors please clarify that?

4. Figure 1b shows that the gap of performance between all classes and the minor class (around epoch 200) is almost the same for all the algorithms. The training accuracy is also very similar for all classes and the minor class from Figure 1a. Does it mean that overfitting to minor classes is not resolved in graphENS? On a similar note, the gap between blue bar and red bar is almost the same for all the algorithms in Figure 1c. However, GraphENS is doing really well to minimize the gap between blue bar and red bar for the neighbor-replacing experiment in Figure 1d. Can authors comment on this and throw some light on the actual reason for the superior performance of graphENS.

**Summary Of The Paper:**

This paper addresses the class imbalance problem for node classification in a graph and points out some issues faced by existing GNNs and  methods to address the same. It nicely depicts the problem of overfitting to minor classes and neighborhood memorization problem for class imbalance through experiments. It proposes n approach GraphENS which can work with any message passing GNN to resolve the problem of class imbalance for node classification. Experimental results show the merit of the proposed algorithm working with multiple GNNs (GCN, GraphSAGE and GAT) on both synthetically imbalanced dataset and real-world imbalanced dataset for node classification.

**Summary Of The Review:**

I am still voting for acceptance since the problem motivation and formulation is done very nicely and experimental section is also strong. The proposed algorithm is simple and intuitive, but not that rigorous.

---

> ### Author Response · Authors · 2021-11-16
> **Response to Reviewer 6ykZ (3 / 3)**
>
> ### [Q4. Figure 1 (b) seems that overfitting to minor classes is not resolved in GraphENS. Why does GraphENS minimize the gap more substantially in neighbor-replacing experiments than in node-replacing experiments?]
>
> For the case of gap between test accuracy of all classes and minor classes (Figure 1 (b)), GraphENS, in fact, shows a decreased gap compared to other baselines- GraphENS:43.40%, OS:52.29%, RW:52.09%  (though it looks similar to other gaps in the graph).
>
> To clearly present alleviation of overfitting to minor classes, we provide the **test accuracy curves for each class** in our revision (**Figure 4 in Appendix A.1**). Note that PubMed-*LT* has three classes and the number of nodes for each class is as follows: (Minor_1: 72, Minor_2: 726, Major:7260).
> In Figure 4 (a), while the training accuracies for minor classes (Minor_1, Minor_2) are highly similar over all methods, GraphENS exhibits significantly superior test accuracies for minor classes (Figure 4 (b,c)). Albeit our method somewhat sacrifices the test accuracy of the major class, we confirmed that our approach substantially mitigates overfitting to the minor classes.
>
> Please note that while Figure 1 (a,b) are experiments showing the existence of overfitting for minor classes, Figure 1 (c,d) are experiments to confirm where this overfitting mainly originates. As the reviewer mentioned, there is a much larger gap in Figure 1(c) than in Figure 1(d), which is precisely our conclusion that this overfitting for minor classes is mainly due to the neighbor memorization problem. In addition, as the reviewer also mentioned, our GraphENS resolves the neighbor memorization problem well in Figure 1(d), and this is the main reason for the superior performance of GraphENS.

---

> ### Author Response · Authors · 2021-11-16
> **Response to Reviewer 6ykZ (2 / 3)**
>
> ### [Q2. The performance of GraphENS on the original citation datasets using GCN, GAT, and GraphSAGE.]
>
> As the reviewer confirmed, our method shows its effectiveness on the manually imbalanced citation datasets in Table 1. It is worth noting that we also evaluate our method on intact citation networks in a semi-supervised learning setting (Table 4). In this experiment, we construct a class-imbalance situation by adjusting the number of labeled nodes without manipulation on the graph.
>
> To further clarify the reviewer’s concerns, we test our method on the original citation networks using entire training nodes as labeled nodes. As the imbalance ratio is low in the original citation networks, our method generates much fewer minor ego networks than class-imbalance scenarios. Nevertheless, our proposed method still brings performance improvements on the original citation networks on Cora and CiteSeer. For PubMed, since the imbalance ratio is 1.90, which is smaller than Cora (3.92) and CiteSeer (2.35), our method is comparable to vanilla GNN models.
>
> | **Method** | **Cora** | **CiteSeer** | **PubMed** |
> |---|:---:|:---:|:---:|
> | (Arch:GCN) | bAcc. \| F1 | bAcc. \| F1 | bACC. \| F1 |
> | Vanilla | 83.00 \| 83.19 | 73.48 \| 73.59 | 87.31 \| 87.29 |
> | GraphENS | **85.11 \| 84.77** | **74.90 \| 74.97** | **87.95 \| 87.34** |
>
> | **Method** | **Cora** | **CiteSeer** | **PubMed** |
> |---|:---:|:---:|:---:|
> | (Arch:GAT) | bAcc. \| F1 | bAcc. \| F1 | bACC. \| F1 |
> | Vanilla | 84.61 \| 84.91 | 72.53 \| 72.15 | 87.60 \| **87.76** |
> | GraphENS | **86.13 \| 85.39** | **74.10 \| 74.00** | **87.74** \| 87.50 |
>
> | **Method** | **Cora** | **CiteSeer** | **PubMed** |
> |---|:---:|:---:|:---:|
> | (Arch:SAGE) | bAcc. \| F1 | bAcc. \| F1 | bACC. \| F1 |
> | Vanilla | 82.69 \| 83.36 | 72.11 \| 71.82 | 88.99 \| **89.13** |
> | GraphENS | **85.29 \| 84.78** | **72.71 \| 72.86** | **89.42** \| 89.06 |
>
> ---
>
> ### [Q3. What is the training, validation and test size for supervised and semi-supervised learning setup? "Supervised Learning" task is also a semi-supervised one. Can authors please clarify that?]
>
> To be clear, we provide the size of training, validation, and test set in all experimental settings on the below table.
>
> | **Dataset**(# of nodes) | **Train set** | **Valid set** | **Test set** | **Unlabeled set** |
> |---|:---:|:---:|:---:|:---:|
> | Cora-*Semi* | 86 | 500 | 1000 | 1122 |
> | CiteSeer-*Semi* | 66 | 500 | 1000 | 1761 |
> | PubMed-*Semi* | 24 | 500 | 1000 | 18193 |
> | Cora-*LT* | 631 | 500 | 1000 | 0 |
> | CiteSeer-*LT* | 611 | 500 | 1000 | 0 |
> | PubMed-*LT* | 8058 | 500 | 1000 | 0 |
> | AmazonPhoto | 5162 | 1000 | 1488 | 0 |
> | AmazonComputers | 11042 | 1090 | 1620 | 0 |
>
> In comparison to the ‘Semi-Supervised Learning’ experiment, ‘Supervised Learning’ setting has no unlabeled nodes in the training set. The purpose of our experiments is to verify the ability to handle not semi-supervised learning, but class-imbalance in node classification. Thus, to validate our algorithms while minimizing the effect of unlabeled nodes, we design “supervised learning settings”, which do NOT have unlabeled nodes in the training set. Compared to the semi-supervised learning setting, the portion of labeled nodes over entire nodes in the supervised learning setting is much larger. To demonstrate the effectiveness of our approach, we evaluate our method on both supervised learning settings (Table 1) and semi-supervised settings with public split (Table 4).

---

> > ### Comment · Reviewer_6ykZ · 2021-11-26
> > **Further Doubts**
> >
> > In Q2. Reply, what is the difference between the three tables? Should architectures be different there?
> >
> > In Q3. Reply, "‘Supervised Learning’ setting has no unlabeled nodes in the training set." - I assume you use the original graph (structure + attributes) as input to your algorithm, while vary the amount of node labels used during the training. In that case, even test nodes are also present in the training but they are still unlabeled. "Training set" by default means examples with labels. So the statement "no unlabeled nodes in the training set" is not clear to me.

---

> > > ### Author Response · Authors · 2021-11-26
> > > **Second Response to Reviewer 6ykZ**
> > >
> > > Thank you for checking out our response. We address the reviewer’s further comments in the below.
> > >
> > > ### [In Q2. Reply, what is the difference between the three tables? Should architectures be different there?]
> > >
> > > We apologize for the typos in our tables. Each table represents the result of each GNN architecture (**GCN**, **GAT**, and **SAGE)**. We corrected the architecture name in the tables. Thank you for pointing it out.
> > >
> > > ---
> > >
> > > ### [In Q3. Reply, test nodes are also present in the training but they are still unlabeled. "Training set" by default means examples with labels. So, the statement “no unlabeled nodes in the training set” is not clear to me.]
> > >
> > > For the semi-supervised learning setting, we follow the public split of train/validation/test nodes [1]. In this setting, only nodes of the train split are labeled in the training phase. Unlabeled nodes which do not belong to this train split are intact and connected to the labeled nodes.
> > >
> > > In the supervised learning setting, we keep the public split of validation/test nodes unmodified. However, from the nodes that do not belong to the validation/test split (labeled training + unlabeled training nodes), we remove them and corresponding edges from the original graph to make the citation graph follow a long-tailed distribution [2]. Then, we assign the labels to the remaining nodes (even for originally unlabeled training nodes). We designed this setting to purely focus on the ability to handle the highly class-imbalance problem while reducing the effects of unlabeled nodes. Thus, the following statement would be more precise: “unlabeled nodes are no longer present in the modified graph except for validation/test nodes.”
> > >
> > > As the reviewer mentioned, it is hard to state that our setting is a rigorously “supervised learning setting” in that the nodes of public validation/test split are still utilized as unlabeled nodes to compute the embeddings of labeled train nodes (transductive setting). We will rectify the description and title of the setting following the reviewer’s comments.
> > >
> > > For information only, these validation/test nodes are NOT used as minor or target nodes in our algorithm.
> > >
> > > Thank you again for your thoughtful feedback.
> > >
> > >
> > > [1] Zhilin Yang et al., Revisiting semi-supervised learning with graph embeddings, ICML 2016, Workshop.
> > >
> > > [2] Cui, Yin, et al. "Class-balanced loss based on effective number of samples." CVPR 2019.

---

> ### Author Response · Authors · 2021-11-16
> **Response to Reviewer 6ykZ (1 / 3)**
>
> We sincerely thank you for your constructive and thoughtful feedback.
>
> ### [Q1. The proposed algorithm GraphENS is very heuristic in nature. It would have been great if some objective / cost function can be formulated and these steps could have been deduced by solving that. At least, authors should give more justification on their need for node classification.]
>
> As with numerous mixup-like augmentations in other classification tasks, it is hard to explicitly formulate how our method directly resolves the objective. However, the rationales of our algorithm for node classification can be explained in terms of two points - expansion and smoothness of decision boundary as other researchers did.
>
> Our method aims to enlarge and smooth the decision boundary of the minor class. First, to extend the boundary of the minor class, GraphENS utilizes the manifold assumption that “similar predictions of neural networks indicate the close proximity in the manifold” [1,2] as a key inductive bias. By synthesizing the minor ego network using the target node of the entire class, our method widens the decision boundary effectively in that we interpolate the ego networks based on prediction similarity. In the vision domain, M2m[3], one of the imbalance-handling methods, also translates major class data into minor class to prevent overfitting on minor classes exploiting our assumption implicitly. This extrapolation scheme for the minor class becomes more significant when the number of data is highly scarce.
> To verify the impact of this extrapolation strategy, we conduct comparative experiments with GraphSMOTE, which only mixes instances of the identical minor class. Although extreme, to highlight the benefit of the extrapolation strategy, we intentionally test our method and GraphSMOTE on extreme cases where major classes have only four labeled nodes.
> In the below table, we confirmed that our method outperforms GraphSMOTE and the performance gaps get larger than the standard case.
>
> | **Method** | **Cora-*Semi*** | **CiteSeer-*Semi*** | **PubMed-*Semi*** |
> |---|:---:|:---:|:---:|
> | (Arch:GCN, Imbalance ratio:2) | standard F1 \| extreme F1 | standard F1 \| extreme F1 | standard F1 \| extreme F1 |
> | GraphSMOTE | 73.94 \| 58.84 | 62.71 \| 39.55 | 73.45 \| 70.85 |
> | GraphENS | **74.97 \| 61.72** | **63.06 \| 48.53** | **74.44 \| 71.96** |
>
> Another advantage of our method for node classification is that the generalization performance of GNNs can be improved by smoothing the decision boundary. As investigated in [4], convex combinations of node features encourage the smoothness of the boundary. We will include the visualized decision boundary of minor classes using 2D bottleneck hidden representations of GNN in our revision. Therefore, our method could construct an unbiased estimator to the label distribution of the training set by enlarging the decision boundary of minor classes smoothly.
>
> [1] Van Engelen, et al. "A survey on semi-supervised learning." Machine Learning 109.2 (2020): 373-440.
>
> [2] Iscen, Ahmet, et al. "Label propagation for deep semi-supervised learning." CVPR 2019.
>
> [3] Kim, Jaehyung, et al. "M2m: Imbalanced classification via major-to-minor translation." CVPR 2020.
>
> [4] Verma, Vikas, et al. “Manifold mixup: Better representations by interpolating hidden states” ICML 2019.

---

> ### Comment · Reviewer_6ykZ · 2021-11-26
> **Author response addresses most of my concerns**
>
> Authors are able to address most of my concerns, except the first one about the heuristic nature of the proposed algorithm. However, I agree that proposing a formal method seems to be highly non-trivial and this does not affect other findings in the paper. I thank the authors for the response and clarifications.

---

### Official Review · Reviewer_KQwg · 2021-11-05

**Correctness:** 3
**Technical Novelty And Significance:** 2
**Empirical Novelty And Significance:** 3
**Recommendation:** 6
**Confidence:** 4

**Main Review:**

Representation learning for node classification usually has class-imbalance issue. Some labels might cover a few number nodes. This work empirically indicates out that the learned node representations will be influenced by the neighbor label distribution. In particular, those nodes with minor class can have the neighbor memorization problem, that is, the target node's embedding might mainly represent the neighbor nodes but not itself. Empirical analysis in this work supports this claim clearly. Then a data augmentation method based on this observation is proposed. Experiments with class-imbalance setting are conducted to demonstrate the superiority of the proposed augmentation method.

Concerns:
 1. The notations in this work are not easy to follow. Authors use many kinds of symbols like minor, target, seen, or unseen to represent the types of nodes. However, it's very difficult to tell their difference in the manuscript. Could you please explain what's the target node and its difference to the nodes with class? Should the minor nodes be the target nodes? Please check the manuscript again and consider to present the types of nodes in alternative way.

2. It's difficult to draw connection between the proposed method and the solution to address the neighbor memorization problem. We can only see an augmented data by mixing up the node feature and neighbor nodes. Could please explain how to created virtual label for the synthesized node, and why the augmentation method can solve the neighbor memorization problem?

3. Could you please consider baselines which can deal with heterophily network [1]? The neighbor memorization seems to be similar with the definition of heterophily, where the target usually has different labels from the neighbors.

References:
[1] Jiong Zhu et al., Graph neural networks with heterophily, AAAI 2021.




**Summary Of The Paper:**

This paper exploring neighborhood memorization problem and proposes a neighbor-aware data augmentation method for node classification task.

**Summary Of The Review:**

This work studies an interesting problem, but the presentation is not clear expressing the motivation.

---

> ### Author Response · Authors · 2021-11-16
> **Response to Reviewer KQwg (2 / 2)**
>
> ### [Q3. Could you please consider baselines which can deal with heterophily network [1]?  The neighbor memorization seems to be similar with the definition of heterophily.]
>
> We thank the reviewer for informing us about this paper. This work proposed a framework called CPGNN [1] that can estimate the class-pair heterophily in the graph by using a learnable compatibility matrix. Although modeling the heterophily of a graph enables to adaptively regulate the aggregation of the neighbor information, it is not directly related to neighbor memorization problems.
> To show the effectiveness of our approach further, we compare our method with CPGNN in an imbalanced semi-supervised learning setting. Our method significantly outperforms CPGNN, which estimates the compatibility matrix and utilizes it in label propagation, over three benchmark datasets. We conjecture that CPGNN shows inferior performance since the estimation error of the compatibility matrix becomes large due to the limited number of minor class nodes. Note that we used the released code for comparison and selected the best model based on validation accuracy among CPGNN-Cheby-1, and CPGNN-Cheby-2. We will add this result in the revision.
>
> | **Method** | **Cora-*Semi*** | **CiteSeer-*Semi*** | **PubMed-*Semi*** |
> |---|:---:|:---:|:---:|
> | (Arch:GCN) | bAcc. \| F1 | bAcc. \| F1 | bAcc. \| F1 |
> | CPGNN | 58.30 \| 54.79 | 41.47 \| 34.85 | 68.41 \| 60.56 |
> | GraphENS | **65.01 \| 64.69** | **49.55 \| 46.27** | **70.68 \| 68.19** |
>
> [1] Jiong Zhu et al., Graph neural networks with heterophily, AAAI 2021.

---

> ### Author Response · Authors · 2021-11-16
> **Response to Reviewer KQwg (1 / 2)**
>
> We thank the reviewer for the helpful and insightful comments. We address the reviewer’s concerns below.
>
> ### [Q1. Could you please explain what’s the target node and its difference to the nodes with class?]
>
> We first apologize for the ambiguous notations. We will keep your comments in mind and carefully revise the notations of node types in alternative ways for better presentation in our final revision.
>
> Meanwhile, we here reiterate our notations below for you:
> The key concept of our method is to generate an ego network for the minor class using two nodes: $v_{minor}$ and $v_{target}$. Node features of these two nodes are mixed to determine a central node of the ego network via convex combination. $v_{minor}$ is the node sampled from the minor class to be augmented (oversampled). That is, the central node of a synthesized ego network acquires an identical label of $v_{minor}$.
> On the other hand, there is no restriction for $v_{target}$: $v_{target}$ is sampled from the entire class and exploited to synthesize the ego network of generated/augmented node.
> Thus, the target node $v_{target}$ does not have to be a minor node. Detailed justification of this strategy for target node selection is discussed in Section 5.2 and our answers to Q2 below.
>
> ---
>
> ### [Q2.  How to create a virtual label for the synthesized node, and why the augmentation method can solve the neighbor memorization problem?]
>
> As briefly mentioned in Q1, the virtual minor node with a mixed node feature gets an identical one-hot label with the $v_{minor}$ (not the soft label).
>
> To resolve neighbor memorization issues, GraphENS aims to generate virtual minor nodes with diverse but probable neighbor sets. The neighbor set for a virtual minor node is determined by a mixed adjacent distribution of $\mathcal{N}(v_{minor})$ and $\mathcal{N}(v_{target})$. To simulate a more diverse neighborhood structure, we do NOT restrict $v_{target}$ to the identical minor class of $v_{minor}$.
>
> However, simply assigning diverse neighbors to a virtual minor node can give the harmful supervisory signal to learning GNNs since the virtual minor node has an identical label with the $v_{minor}$. In this context, our method is designed to effectively restrain the generation of noisy neighbor sets by exploiting prediction similarity.
> KL divergence between the predictions of $v_{minor}$ and $v_{target}$ is exploited in mixing adjacent distributions. We also mask the target node feature proportional to KL divergence to filter-out class-specific features of $v_{target}$. From these design choices, we can generate reliable ego networks with diverse neighbor sets.
>
> Of course, we can simply reduce the noisy signal by mixing the ground-truth label of $v_{minor}$ and $v_{target}$. However, this soft label strategy is not sufficient to alleviate the bias to major classes under a class imbalance scenario. If we adopt a label mixing scheme, the performance of our method degrades significantly (below table). We thus believe that conventional label mixing is not appropriate to handle class-imbalance problems.
>
> Please let us know if there were any unasked questions or unclear parts that we have not addressed up to your satisfaction.
>
> | **Method** | **Cora-*LT*** | **CiteSeer-*LT*** | **PubMed-*LT*** |
> |---|:---:|:---:|:---:|
> | (Arch:GCN) | bAcc. \| F1 | bAcc. \| F1 | bAcc. \| F1 |
> | GraphENS (*w/ label mix*) | 68.18 \| 70.37 | 52.04 \| 50.02 | 70.36 \| 71.47 |
> | GraphENS | **72.94 \| 73.13** | **60.19 \| 58.67** | **74.13 \| 74.58** |

---

> > ### Comment · Reviewer_KQwg · 2021-12-03
> > **Thanks to authors for the rebuttal**
> >
> > I really appreciate the informative response from the authors. Basically I like the idea very much. Parts of my concerns have been addressed well. Before changing my score, I still need authors to clarify following points:
> >
> > 1. The setting of replacement experiments designed for demonstrating neighbor memorization issue. Is the replacement only applied to a learned GNN or the replaced nodes will be trained again after the replacement?
> >
> > 2.  Does "the seen, unseen, and anchor nodes from the identical minor classes" mean that they have the same node label?
> >
> > 3. The fairness of replacement experiments for the seen nodes. It's noted that the "seen" node replaces the "unseen" node for studying neighbor memorization problem. In this case, the neighbors of the unseen (nodes from the test set) nodes might be not trained at all. Is it possible that the accuracy downgrades is affected by this situation? We can see that classification accuracy for seen nodes with the neighbors (from the training set) of anchor nodes is stable for both two kinds of replace experiments.

---

> > > ### Author Response · Authors · 2021-12-03
> > > **Second Response to Reviewer KQwg**
> > >
> > > We sincerely thank the reviewer for checking out our response and assessing our idea to be interesting.
> > >
> > > ### [Q1.The setting of replacement experiments designed for demonstrating neighbor memorization issue. Is the replacement only applied to a learned GNN or the replaced nodes will be trained again after the replacement?]
> > >
> > > Replacing experiments are applied only in **the inference phase** to estimate how much a learned GNN depends on ‘self node features’ or ‘neighbor structures’. Note that the intention of these experiments is to measure the generalization performance of **trained** GNNs over unseen self node features in node-replacing experiments or over unseen neighbor structures in neighbor-replacing experiments.
> > >
> > > ---
> > >
> > > ### [Q2. Does "the seen, unseen, and anchor nodes from the identical minor classes" mean that they have the same node label?]
> > >
> > > As the reviewer mentioned, the seen, unseen, and anchor nodes have **the same node label**. The only difference between the three nodes is that seen and anchor nodes are sampled from the **training set** and the unseen node is sampled from the **test set**.
> > >
> > > ---
> > >
> > > ### [Q3. It's noted that the "seen" node replaces the "unseen" node for studying neighbor memorization problem. In this case, the neighbors of the unseen (nodes from the test set) nodes might be not trained at all. Is it possible that the accuracy downgrades is affected by this situation? We can see that classification accuracy for seen nodes with the neighbors (from the training set) of anchor nodes is stable for both two kinds of replace experiments.]
> > >
> > > As the reviewer mentioned, replacing seen ‘self node feature’ or seen ‘neighbor set’ with their **unseen** counterparts would inevitably bring performance degradation since they are not present in the training phase. As such, we **intentionally** measure the performance degradation (sensitivity) in replacing experiments to find out the major factor contributing to this degradation.
> > >
> > > Our key finding is that the performance degradation of neighbor replacing is more significant than that of node replacing, not the performance degradation itself when replacing with unseen.
> > >
> > > We here reference our analysis in Section 3 to clarify our statements:
> > >
> > > “We demonstrate that conventional imbalance handling approaches severely suffer from a neighbor memorization problem in Figure 1(c) and 1(d). The performance drop of conventional algorithms in the neighbor-replacing experiment is almost two times steeper than in the node-replacing setting. These results imply that the neighbor memorization problem is a critical obstacle in properly handling the class-imbalanced problem in node classification tasks.”
> > >
> > > Please let us know if the concern mentioned by the reviewer is not about this.
> > >
> > >
> > > ---
> > >
> > > We will carefully refine our manuscript to clarify the parts mentioned by the reviewer. We sincerely appreciate the reviewer again for the thoughtful comments.
> > >
> > > Please let us know if there were any unasked questions or comments that we have not addressed up to the reviewer’s satisfaction.

---

> > > > ### Comment · Reviewer_KQwg · 2021-12-06
> > > > **Update my rate**
> > > >
> > > > Thank you so much for your response. My concerns have been addressed well. I'd like to upgrade my rate.

---

### Public Comment · ~Joonhyung_Park1 · 2022-08-18
**Official Code of GraphENS**

Our official PyTorch code is available at https://github.com/JoonHyung-Park/GraphENS.

---

### Decision · Program_Chairs · 2022-01-20

**Decision:**

Accept (Poster)

**Comment:**

Although reviews were initially a little polarized, they trend toward accepting the paper after rebuttal and discussion.

The most negative review raised issues of datasets, baselines, and experiments, and various details that they find confusing. These concerns were not shared by the other reviewers for the most part. Following a detailed rebuttal the most negative reviewer ended up siding with the more positive reviewers.